# Ductal or Ngn3$^+$ cells do not contribute to adult pancreatic islet beta-cell neogenesis in homeostasis

Xiuzhen Huang[1,6], Huan Zhao [ID][1,6 ✉], Hui Chen [ID][1,6], Zixin Liu[1], Kuo Liu[2], Zan Lv[1], Xiuxiu Liu[1], Ximeng Han[1], Maoying Han [ID][1], Jie Lu [ID][3], Qiao Zhou[4] & Bin Zhou [ID][1,2,5 ✉]

## Abstract

The adult pancreatic ducts have long been proposed to contain rare progenitors, some of which expressing Ngn3, that generate new beta cells in endocrine-islet homeostasis. Due to their postulated rarity and the lack of definitive markers, the existence or absence of ductal endocrine progenitors remains unsettled despite many studies. Genetic lineage tracing of ductal cells or Ngn3$^+$ cells with currently available CreER drivers has been complicated by off-target labeling of pre-existing beta cells. Here, using dual-recombinase-mediated intersectional genetic strategy and newly-derived *Ngn3-2A-CreER* and *Hnf1b-2A-CreER* knock-in drivers, we succeeded in specifically labeling Ngn3-positive cells and Hnf1b-positive ductal cells without marking pre-existing beta cells. These data revealed no evidence of de novo generation of insulin-producing beta cells from ductal cells or endogenous Ngn3-positive cells in the adult pancreas during homeostasis.

**Keywords** Beta Cell Neogenesis; Ngn3; Pancreatic Ductal Cell; Progenitor
**Subject Categories** Development; Stem Cells & Regenerative Medicine

## Introduction

A major cause of type 1 diabetes and a subset of type 2 diabetes is the deficiency of functional beta cells in the pancreas (Aguayo-Mazzucato and Bonner-Weir, 2018; Zhou and Melton, 2018). Understanding potential sources of new beta cells in the adult pancreas will facilitate both basic and clinical research and inform the treatment of insulin-dependent diabetes. Numerous studies have reported on two major replenishing sources of the beta cell mass: self-replication of pre-existing beta cells and beta cell neogenesis (generation of new beta cells from progenitors or non-beta cells without genetic manipulation) (Zhou and Melton, 2018). The self-replication model was first proved by Dor et al (Dor et al, 2004), with other research groups subsequently coming to the same conclusion using genetic tools (Blaine et al, 2010; Cox et al, 2016; Nir et al, 2007; Perez-Frances et al, 2022; Rankin et al, 2013; Xiao et al, 2013a; Zhao et al, 2021a) or chemical labeling (Teta et al, 2007). The neogenesis model often invokes the existence of progenitor cells expressing Neurogenin 3 (Neurog3, also known as Ngn3), a master regulator of endocrine specification, in extra-islet locations such as the pancreatic ducts (Gradwohl et al, 2000; Gribben et al, 2021; Gribben et al, 2023; Gu et al, 2002; Magenheim et al, 2023; Seymour et al, 2008; Solar et al, 2009; Xu et al, 2008; Yu et al, 2016; Zhou et al, 2007). In 2008, Xu et al utilized a transgenic *Nng3-nLacZ* mouse line and reported for the first time that adult pancreatic Ngn3$^+$ progenitors could contribute to the generation of beta cells after partial duct ligation (PDL) injury (Xu et al, 2008). Other studies, however, showed that *Ngn3* mRNA and protein could be detected in pre-existing beta cells (Wang et al, 2009), and although PDL injury could activate *Ngn3* expression in pancreatic ducts, it does not lead to beta cell neogenesis from duct cells (Kopp et al, 2011; Xiao et al, 2013b).

In addition to Ngn3$^+$ cells, genetic lineage tracing was used extensively to detect endocrine differentiation from pancreatic ducts. Evidence for and against ductal endocrine progenitors was put forward by different groups (Furuyama et al, 2011; Inada et al, 2008; Kopinke et al, 2011; Kopinke and Murtaugh, 2010; Kopp et al, 2011; Solar et al, 2009), with the most recent being Gribben et al reporting a population of ductal Ngn3$^+$ progenitors making a sizable contribution to beta cell mass in homeostasis (Gribben et al, 2021). This report sparked a vigorous debate (Bonner-Weir, 2021; Gribben et al, 2023; Magenheim et al, 2023; Zhao et al, 2021b). In a Matters Arising, Magenheim et al provided new data to challenge this conclusion (Magenheim et al, 2023).

In our view, much of the uncertainty about ductal- or exocrine-resident endocrine progenitors can be traced to deficiencies in the genetic lineage tracing tools presently available. In particular, *Ngn3-CreER* and *Hnf1b-CreER* (used in tracing pancreatic ductal cells) have "off-target" labeling of pre-existing beta cells. In addition, the modest labeling efficiency in some of the ductal tracing experiments was criticized as potentially missing rare ductal cell populations (Gribben et al, 2023). To resolve these issues, we need genetic tools

[1]New Cornerstone Science Laboratory, State Key Laboratory of Cell Biology, Shanghai Institute of Biochemistry and Cell Biology, Center for Excellence in Molecular Cell Science, Chinese Academy of Sciences, University of Chinese Academy of Sciences, 200031 Shanghai, China. [2]School of Life Science, Hangzhou Institute for Advanced Study, University of Chinese Academy of Sciences, 310024 Hangzhou, China. [3]Department of Gastroenterology, Shanghai General Hospital, Shanghai Jiao Tong University School of Medicine, 200080 Shanghai, China. [4]Division of Regenerative Medicine & Hartman Institute for Organ Regeneration, Department of Medicine, Weill Cornell Medicine, 1300 York Avenue, New York, NY 10065, USA. [5]School of Life Science and Technology, ShanghaiTech University, 100 Haike Road, 201210 Shanghai, China. [6]These authors contributed equally: Xiuzhen Huang, Huan Zhao, Hui Chen. ✉E-mail: zhaohuan@sibs.ac.cn; zhoubin@sibs.ac.cn

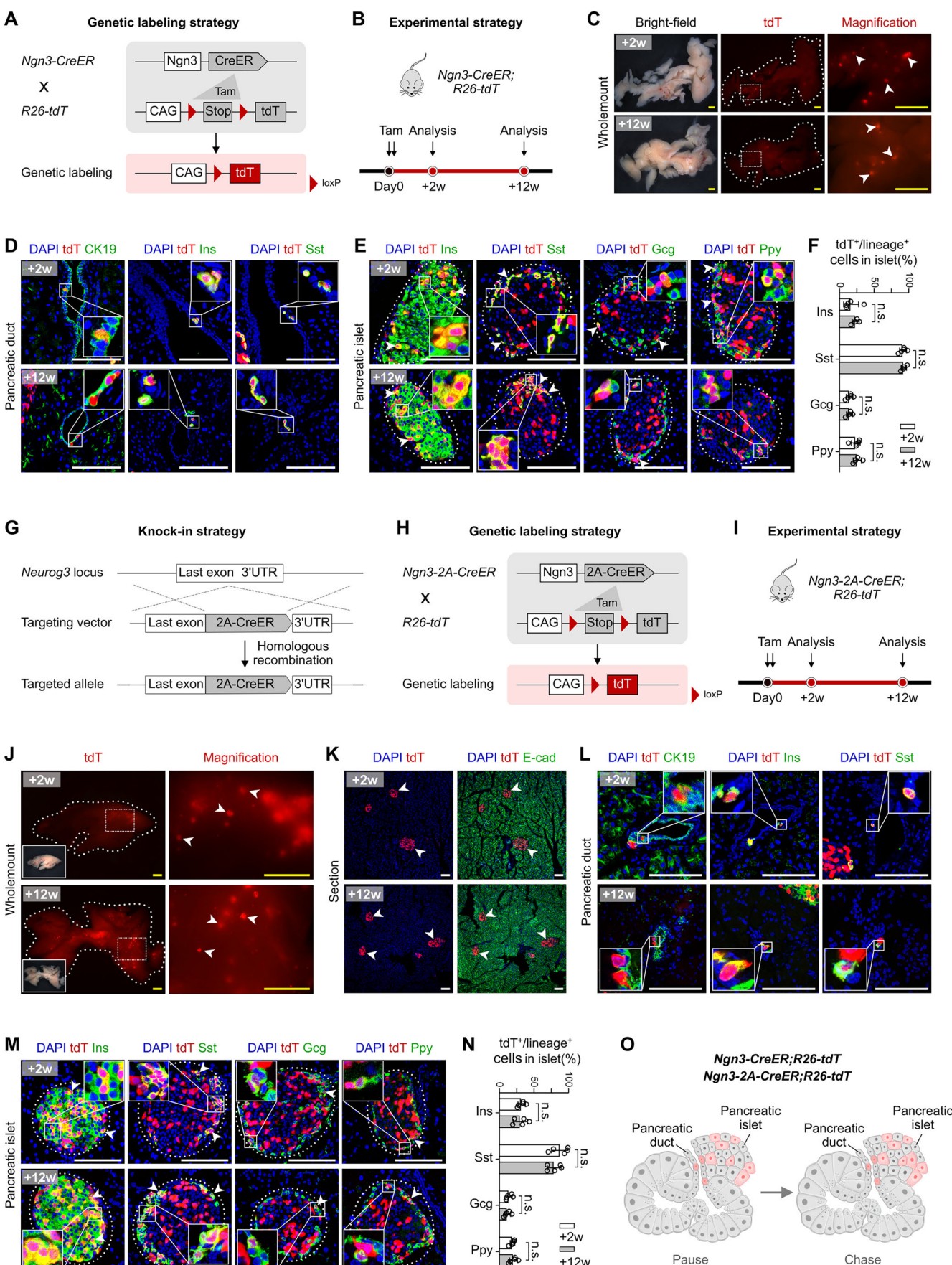

**Figure 1.  Genetic lineage tracing of Ngn3⁺ cells in the pancreas.**

(A) Schematic diagram of the lineage tracing strategy by *Ngn3-CreER;R26-tdT* mice. (B) Schematic showing the experimental strategy. (C) Wholemount fluorescent images of pancreas collected from *Ngn3-CreER;R26-tdT* at the indicated time points after Tam treatment. Arrowheads, tdT⁺ pancreatic islets. (D) Immunostaining for tdT and CK19 or Ins or somatostatin (Sst) on pancreatic sections of *Ngn3-CreER;R26-tdT*. (E) Immunostaining for tdT and Ins or Sst or glucagon (Gcg) or pancreatic polypeptide (Ppy) on pancreatic sections of *Ngn3-CreER;R26-tdT* mice. Arrowheads, tdT⁺lineage⁺ pancreatic endocrine cells. (F) Quantification of the percentage of tdT⁺ cells among pancreatic endocrine cell lineages in the *Ngn3-CreER;R26-tdT* mice. Data are mean ± SD; +2 weeks, $n = 5$ biological replicates; +12 weeks, $n = 4$ biological replicates; Ins: $P = 0.30$; Sst: $P = 0.72$; Gcg: $P = 0.97$; PP: $P = 0.62$; n.s., non-significant. In each sample, islets from 10 pancreas sections were quantified. Two-tailed unpaired Student's *t*-tests were used for statistical comparisons and $P < 0.05$ was accepted as statistically significant. (G) Schematic of the *Ngn3-2A-CreER* knock-in strategy. (H) Schematic of lineage tracing strategy for *Ngn3-2A-CreER;R26-tdT*. (I) Schematic showing the experimental strategy. (J) Wholemount fluorescent images of the pancreas from *Ngn3-2A-CreER;R26-tdT* mice at the indicated time-points after Tam treatment. Arrowheads, tdT⁺ pancreatic islets. (K, L) Immunostaining for tdT and E-cad (K) or CK19 or Ins or Sst (L) on pancreatic sections of *Ngn3-2A-CreER;R26-tdT* mice. Arrowheads in (K), tdT⁺ islets. (M) Immunostaining for tdT and Ins or Sst or Gcg or Ppy on pancreatic sections of *Ngn3-2A-CreER;R26-tdT* Mice. Arrowheads, tdT⁺lineage⁺ pancreatic endocrine cells. (N) Quantification of the percentage of tdT⁺ cells in pancreatic endocrine cell lineages of *Ngn3-2A-CreER;R26-tdT* mice. Data are mean ± SD; +2 weeks, $n = 5$ biological replicates; +12 weeks, $n = 5$ biological replicates; Ins: $P = 0.76$; Sst: $P = 0.26$; Gcg: $P = 0.42$; PP: $P = 0.67$; n.s., non-significant. In each sample, islets from 10 pancreas sections were quantified. Two-tailed unpaired Student's *t* tests were used for statistical comparisons and $P < 0.05$ was accepted as statistically significant. (O) Cartoon image showing that *Ngn3-CreER* and *Ngn3-2A-CreER* label a subset of ductal cells and endocrine cells in the adult pancreas. Scale bars, 1 mm (yellow) and 100 μm (white). Each image is representative of 4–5 individual mouse samples. See also Fig. EV1. Source data are available online for this figure.

to mark ducts at exceptionally high efficiency without off-target labeling. Here, we generated new knock-in mouse lines *Ngn3-2A-CreER* and *Hnf1b-2A-CreER*, which do not affect endogenous gene expression and are highly efficient in the pancreas. Using a dual-recombinase-mediated intersectional genetic approach, we systematically investigated whether Ngn3-expressing progenitors and pancreatic ductal epithelial cells contribute to beta cell neogenesis during homeostasis in adults. Our results unambiguously support the contention that Nng3-expressing progenitors and pancreatic ductal cells do not generate beta cells de novo in adult pancreas homeostasis.

## Results

### Ngn3-CreER and Ngn3-2A-CreER label ductal cells and also various islet cells

The *Ngn3-CreER* transgenic line used in Gribben et al (Gribben et al, 2021) appears to label a population of pre-existing beta cells in addition to rare ductal cells. Magenheim and colleagues (Magenheim et al, 2023) further analyzed a *Ngn3-CreER* knockin line that labels islet but not ductal cells. The question arises as to exactly what cell lineages *Ngn3-CreER* should be labeling that faithfully recapitulate its endogenous expression.

We first evaluated the transgenic mouse line *Ngn3-CreER* (Gu et al, 2002) as Gribben et al reported (Gribben et al, 2021), and crossed it with *R26-tdT* (*R26-loxP-Stop-loxP-tdTomato*) (Madisen et al, 2010) mice to generate *Ngn3-CreER;R26-tdT* for pulse-and-chase experiments (Fig. 1A). We treated *Ngn3-CreER;R26-tdT* with Tam (tamoxifen) at the adult stage and collected the pancreases for analysis at 2 weeks and 12 weeks afterwards (Fig. 1B). Wholemount fluorescent images highlighted localized tdT (tdTomato) signals in pancreatic tissues collected at both 2 weeks and 12 weeks after Tam induction (Fig. 1C). To clarify the identities of the tdT⁺ cells, we performed immunostaining for tdT and the pancreatic ductal cell marker CK19 on tissue sections. We found few tdT⁺ cells lining the ductal epithelium, similar to the findings of Gribben et al (Gribben et al, 2021) (Fig. 1D). We also found that the ductal tdT⁺ cells expressed insulin (Ins) and the delta cell marker somatostatin (Sst) (Fig. 1D). Consistent with the findings of Magenheim et al (Magenheim et al, 2023), a substantial proportion of pancreatic endocrine cells (including

beta cells, delta cells, alpha cells, and PP cells) were labeled at both the "pulse" time point (2 weeks post-Tam) and the "chase" time point (12 weeks of post-Tam) in *Ngn3-CreER;R26-tdT* mice (Fig. 1E). Quantitively, 16.86 ± 10.77% of the beta cells were tdT⁺ at the "pulse" time point, and the percentage of pulse-labeled beta cells did not increase significantly after chase period (23.47 ± 5.04% at 12 weeks post-Tam) (Fig. 1F).

The above mouse line, *Ngn3-CreER*, was generated based on random transgene integration (Gu et al, 2002). The *Ngn3-CreER* knockin line used in the Magenheim study inactivated one *Ngn3* allele, which may perturb endogenous *Ngn3* transcription (Magenheim et al, 2023). We therefore decided to create a new mouse line, *Ngn3-2A-CreER* by inserting the CreER into the 3'UTR (3' untranslated region) of the endogenous *Ngn3* gene, preserving endogenous *Ngn3* transcription (Fig. 1G). We crossed *Ngn3-2A-CreER* with the *R26-tdT* reporter mouse line to obtain *Ngn3-2A-CreER;R26-tdT* mice (Fig. 1H) and analyzed the mice at 2 weeks and 12 weeks post-Tam (Fig. 1I). Wholemount imaging of *Ngn3-2A-CreER;R26-tdT* pancreases revealed tdT⁺ signal patterns similar to that of *Ngn3-CreER;R26-tdT* (Fig. 1J). *Ngn3-2A-CreER* labeled scattered ductal cells, the majority of Sst⁺ delta cells in islets (88.08 ± 12.89%), and a sizable population of Ins⁺ beta-cells (32.18 ± 5.34%) at 2 weeks post-Tam (Fig. 1K–M). At 12 weeks post-Tam, the proportion of labeled endocrine cells did not increase significantly over time (Fig. 1N). Without Tam, very few tdT⁺ cells were detected in the islets from *Ngn3-CreER;R26-tdT* (0.04 ± 0.03%), and *Ngn3-2A-CreER;R26-tdT* (0.02 ± 0.02%) (Fig. EV1A–J).

These data demonstrate that the transgenic *Ngn3-CreER* and the knock-in *Ngn3-2A-CreER* both labeled hormone-positive endocrine cells in the ductal epithelium and pancreatic islets (Fig. 1O). The similar labeling percentage of endocrine cells from 2 weeks to 12 weeks post-Tam suggests a lack of substantial de novo proliferation or transdifferentiation by the labeled cells during the pulse-and-chase time window. These data also highlighted difficulties in interpreting the lineage tracing data due to "ectopic" labeling of pre-existing endocrine cells.

### Simultaneous tracing of Ngn3⁺ non-beta cells, Ngn3⁺ beta cells, and Ngn3⁻ beta cells by a traffic light reporter

A dual-recombinase-based lineage tracing strategy that combines the Cre/loxP and Dre/rox systems could allow more specific

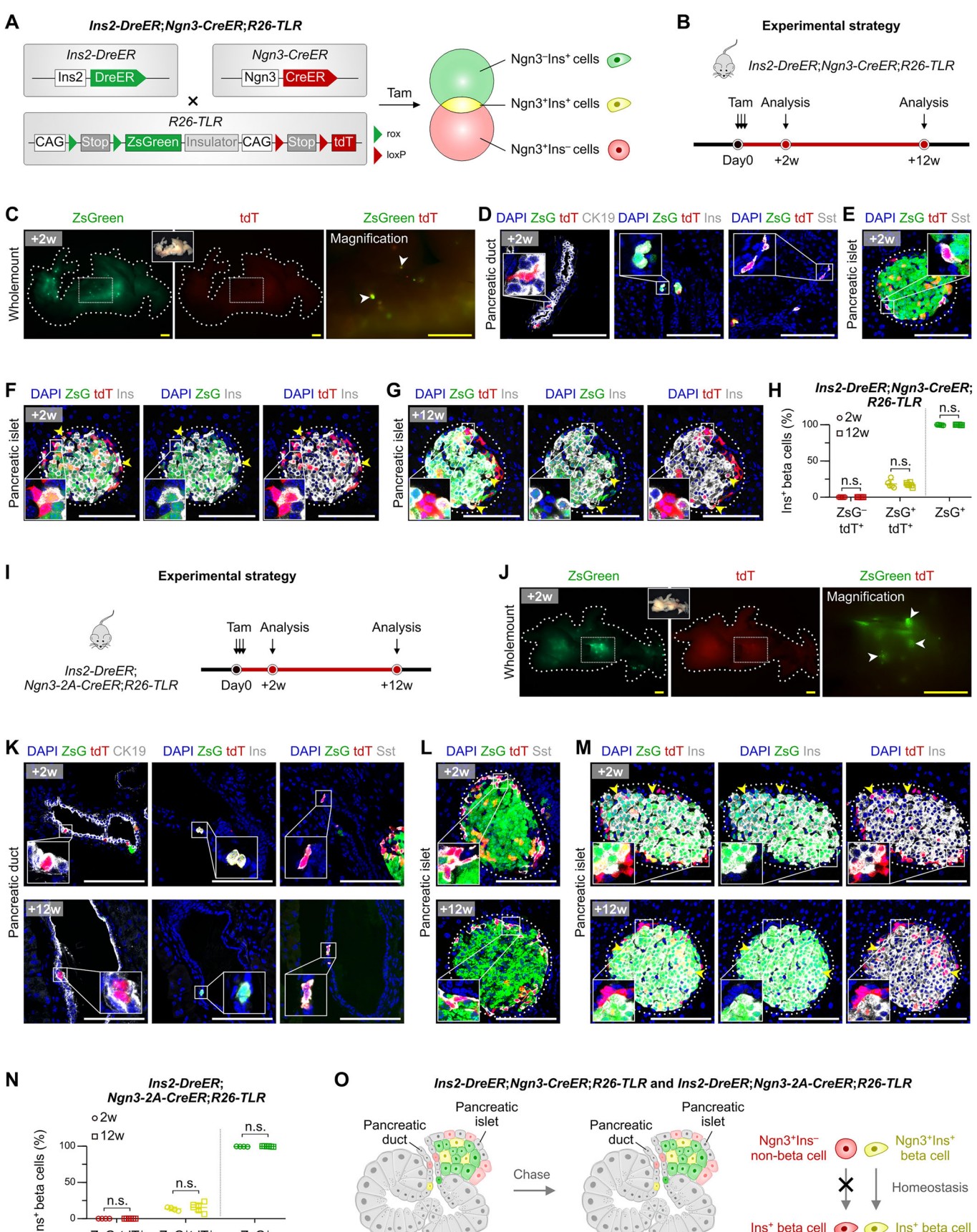

◀

**Figure 2.  Simultaneous tracing of Ngn3+Ins- non-beta cells, Ngn3+Ins+ beta cells, and Ngn3-Ins+ beta cells by a triple light reporter.**

(A) Schematic illustrating the labeling strategy by *Ins2-DreER;Ngn3-CreER;R26-TLR* mice. (B) Schematic showing the experimental strategy. (C) Wholemount fluorescent images of pancreas collected from *Ins2-DreER;Ngn3-CreER;R26-TLR* mice after 2 weeks of Tam treatment. Arrowheads, pancreatic islets. (D) Immunostaining for ZsGreen, tdT, and CK19 or Ins or Sst on pancreatic sections of *Ins2-DreER;Ngn3-CreER;R26-TLR* mice. (E) Immunostaining for ZsGreen, tdT, and Sst on pancreatic sections of *Ins2-DreER;Ngn3-CreER;R26-TLR* mice. (F, G) Immunostaining for ZsGreen, tdT, and Ins on pancreatic sections of *Ins2-DreER;Ngn3-CreER;R26-TLR* mice after 2 weeks (F) and 12 weeks (G) of Tam treatment. Arrowheads, ZsGreen+tdT+Ins+ pancreatic beta cells. (H) Quantification of the percentage of ZsGreen-tdT+ and ZsGreen+tdT+ cells and ZsGreen+ cells in Ins+ beta cells of *Ins2-DreER;Ngn3-CreER;R26-TLR* mice. Data are mean ± SD; +2 weeks, $n = 5$ biological replicates; +12 weeks, $n = 6$ biological replicates; ZsGreen-tdT+: $P = 0.13$; ZsGreen+tdT+: $P = 0.71$; ZsGreen+: $P = 0.51$; n.s., non-significant. In each sample, islets from 10 pancreas sections were quantified. Two-tailed unpaired Student's *t* tests were used for statistical comparisons and $P < 0.05$ was accepted as statistically significant. (I) Schematic diagram showing the experimental strategy. (J) Wholemount fluorescence images of the pancreas from *Ins2-DreER;Ngn3-2A-CreER;R26-TLR* mice after 2 weeks of Tam treatment. Arrowheads, pancreatic islets. (K) Immunostaining for ZsGreen, tdT, and CK19 or Ins or Sst on pancreatic sections of *Ins2-DreER;Ngn3-2A-CreER;R26-TLR* mice after 2 weeks (top) and 12 weeks (bottom) of Tam treatment. (L, M) Immunostaining for ZsGreen, tdT, and Sst (L) or Ins (M) on pancreatic sections of *Ins2-DreER;Ngn3-2A-CreER;R26-TLR* mice after 2 weeks (top) and 12 weeks (bottom) of Tam treatment. Arrowheads, ZsGreen+tdT+Ins+ pancreatic beta cells. (N) Quantification of the percentages of ZsGreen-tdT+ and ZsGreen+tdT+ cells and ZsGreen+ cells in Ins+ beta cells. Data are mean ± SD; +2 weeks, $n = 4$ biological replicates; +12 weeks, $n = 6$ biological replicates; ZsGreen-tdT+: $P = 0.73$; ZsGreen+tdT+: $P = 0.55$; ZsGreen+: $P = 0.36$; n.s., non-significant. In each sample, islets from 10 pancreas sections were quantified. Two-tailed unpaired Student's *t* tests were used for statistical comparisons and $P < 0.05$ was accepted as statistically significant. (O) Cartoon image showing simultaneous tracing of Ngn3+Ins- non-beta cells, Ngn3+Ins+ beta cells, and Ngn3-Ins+ beta cells in the pancreas using *Ins2-DreER;Ngn3-CreER;R26-TLR* and *Ins2-DreER;Ngn3-2A-CreER;R26-TLR* mice. Ngn3+Ins- non-beta cells do not contribute to beta cell neogenesis during homeostasis. Scale bars, 1 mm (yellow) and 100 μm (white). Each image is representative of 4–6 individual mouse samples. See also Fig. EV2. Source data are available online for this figure.

labeling of particular cell lineages (He et al, 2017). We employed the *R26-TLR* (Rosa26-traffic light reporter) mouse line, which has three distinct fluorescent reporters that can trace Cre+Dre+, Cre-Dre+, Cre+Dre- cells simultaneously in one mouse (Liu et al, 2020). We also utilized the highly efficient insulin-specific inducible mouse line *Ins2-DreER*, which labels over 99% of beta cells (Zhao et al, 2021a). We generated *Ins2-DreER;Ngn3-CreER;R26-TLR* mice (Fig. 2A) in which Ins+ beta cells were marked by ZsGreen, Ngn3+Ins- cells by tdT, and Ngn3+Ins+ cells by both ZsGreen and tdT (Fig. 2A). In this design, if Ngn3+Ins- non-beta cells, which contain putative progenitors, give rise to new beta cells, then a subset of beta cells would be marked by tdT+ only after the chase period.

We first validated that nearly all Ins+ beta cells (99.92 ± 0.07%) could be labeled with ZsGreen in *Ins2-DreER;R26-TLR* after Tam injection with high specificity (Fig. EV2A–F). Next, analyzing samples of *Ins2-DreER;Ngn3-CreER;R26-TLR* pancreases 2 weeks after Tam, we readily detected ZsGreen and tdT fluorescence in wholemount preparations (Fig. 2B,C). Combinatorial staining of pancreatic sections with antibodies against ZsGreen, tdT, CK19, Sst, and Ins revealed three distinct cell populations in ducts and islets: tdT+ZsGreen-Ins- non-beta cells, tdT-ZsGreen+Ins+ beta cells, and tdT+ZsGreen+ beta cells (i.e. Ngn3+Ins+ beta cells) (Fig. 2D–F). The vast majority of tdT+ZsGreen- non-beta cells were located in the pancreatic islets, most of which were pancreatic Sst+ delta cells, and approximately single-digit numbers of tdT+ZsGreen- non-beta cells could be found located in the pancreatic ductal epithelium per tissue section, in agreement with the findings of Gribben et al (Gribben et al, 2021) (Fig. 2D–F). 12 weeks post-Tam, we did not detect any Ins+ cells expressing tdT single fluorescence in ducts or in islets (Fig. 2G). Quantitative data showed that none of Ins+ beta cells was labeled with tdT at 2 or 12 weeks after Tam treatment (Fig. 2H, 2 weeks: 0.02 ± 0.01%; 12 weeks: 0.06 ± 0.06%). Moreover, there was no significant difference in the proportion of ZsGreen+tdT+ beta cells at 2 weeks (18.43 ± 5.54%) or at 12 weeks (17.42 ± 2.82%) (Fig. 2H), suggesting a lack of substantial proliferative advantage for the Ngn3+Ins+ beta cell subpopulation during homeostasis. In addition, we found no

detectable dilution of ZsGreen+ beta cells labeled with *Ins2-DreER* from 2 weeks to 12 weeks (99.68 ± 0.31% in 2 weeks and 99.79 ± 0.22% in 12 weeks, Fig. 2H). These data together indicate that Ngn3+ non-beta cells do not generate any detectable new beta cells during homeostasis, consistent with the finding of previous studies (Dor et al, 2004; Perez-Frances et al, 2022; Xiao et al, 2013a; Zhao et al, 2021a).

To validate the above finding, we crossed *Ins2-DreER;R26-TLR* with the knockin *Ngn3-2A-CreER* (Fig. 2I). Wholemount fluorescence imaging of Tam-treated pancreases revealed a patchy distribution of ZsGreen+ islets (Fig. 2J). Combinatorial immunostaining revealed no Ins+ beta cells expressing tdT single fluorescence at either 2 weeks or 12 weeks (Fig. 2J–N, 2 weeks: 0.04 ± 0.03%; 12 weeks: 0.03 ± 0.03%), similar to the results of the *Ins2-DreER;Ngn3-CreER;R26-TLR* system. As controls, both *Ins2-DreER;Ngn3-CreER;R26-TLR* and *Ins2-DreER;Ngn3-2A-CreER;R26-TLR* mice expressed virtually no fluorescent proteins in the absence of Tam (Fig. EV2G–N). The above data demonstrated that Ngn3+Ins- non-beta cells (containing Ngn3+ progenitors) do not give rise to new Ins+ beta cells under homeostatic conditions in adults (Fig. 2O).

## Ngn3+ non-beta cells did not contribute to beta cells during homeostasis in adults

To independently address whether Ngn3+ non-beta cells contribute to beta cells in adults, we adopted another nested reporter, *NR1* (He et al, 2017), in which Cre-based lineage tracing is controlled by Dre/rox recombination. Briefly, this strategy using *Ins2-DreER;Ngn3-CreER;NR1* mice labels Ngn3+ non-beta cells with ZsGreen, whereas all beta cells including Ngn3+ beta cells, are labeled with tdT, as any Cre-activated reporter genes are excised by Dre/rox in beta cells (Fig. 3A). In this design, if Ngn3+ non-beta cells could differentiate into beta cells during the chase period, we would detect ZsGreen+tdT- beta cells at the chase time point but not before (Fig. 3A).

We first examined the labeling efficiency of beta cells using *Ins2-DreER;NR1, and* found that virtually all Ins+ beta cells (99.87 ± 0.21%) could be labeled by Dre/rox recombination

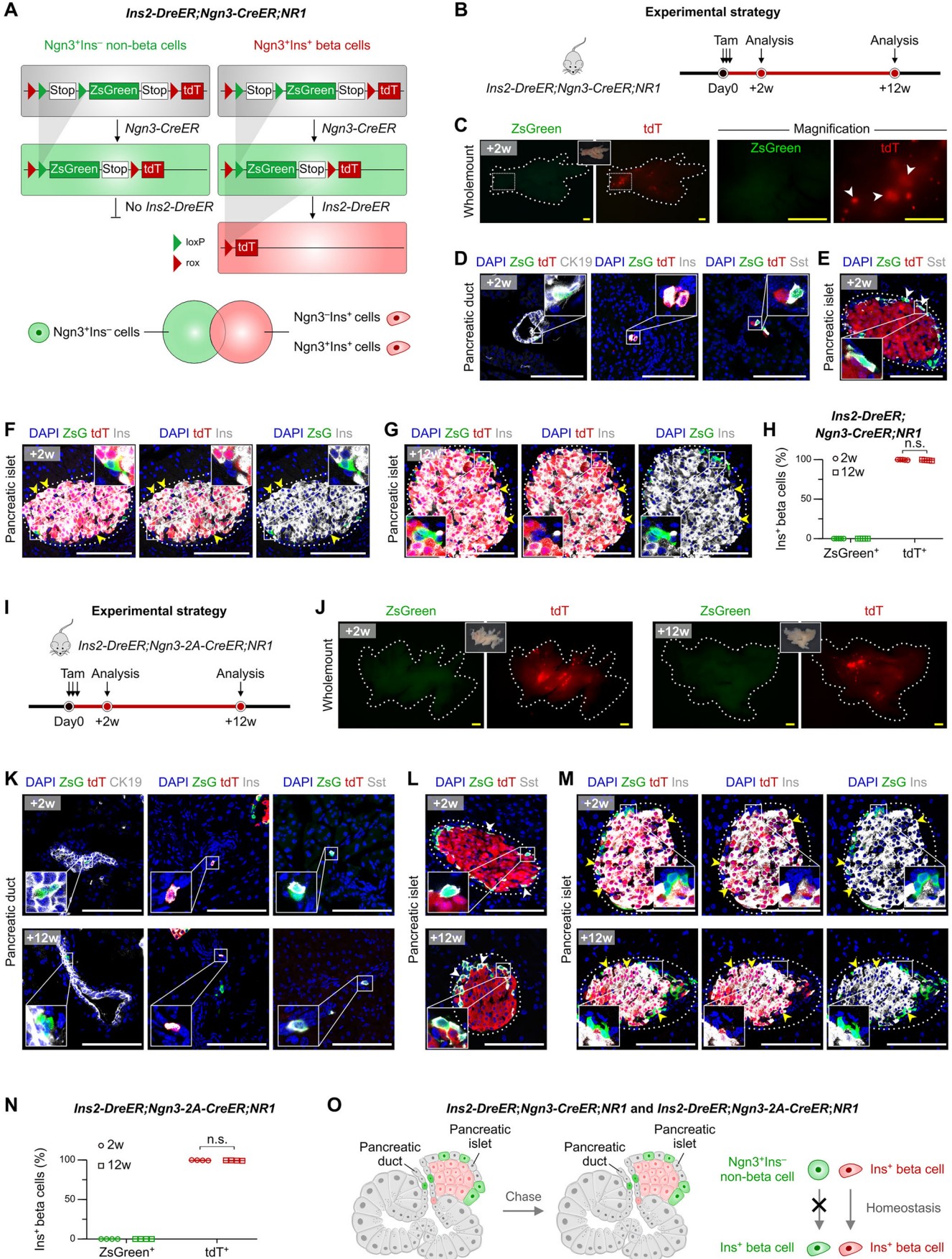

Figure 3. Nested strategy shows Ngn3+Ins− non-beta cells do not contribute to Ins+ beta cells during adult homeostasis.

(A) Schematic of the labeling strategy by *Ins2-DreER;Ngn3-CreER;NR1* mice. (B) Schematic showing the experimental strategy. (C) Wholemount fluorescence image of the pancreas from *Ins2-DreER;Ngn3-CreER;NR1* mice after 2 weeks of Tam treatment. Arrowheads, pancreatic islets. (D) Immunostaining for ZsGreen, tdT, and CK19 or Ins or Sst on pancreatic sections of *Ins2-DreER;Ngn3-CreER;NR1* mice after 2 weeks of Tam treatment. (E) Immunostaining for ZsGreen, tdT, and Sst on pancreatic sections of *Ins2-DreER;Ngn3-CreER;NR1* mice. Arrowheads, ZsGreen+Sst+ pancreatic delta cells. (F, G) Immunostaining for ZsGreen, tdT, and Ins on pancreatic sections of *Ins2-DreER;Ngn3-CreER;NR1* mice after 2 weeks (F) and 12 weeks (G) of Tam treatment. Arrowheads, tdT+Ins+ pancreatic beta cells. (H) Quantification of the percentages of ZsGreen+ or tdT+ cells in Ins+ pancreatic beta cells of *Ins2-DreER;Ngn3-CreER;NR1* mice. Data are mean ± SD; +2 weeks, n = 5 biological replicates; +12 weeks, n = 5 biological replicates; tdT+: $P = 0.14$; n.s., non-significant. In each sample, islets from 10 pancreas sections were quantified. Two-tailed unpaired Student's *t* tests were used for statistical comparisons and $P < 0.05$ was accepted as statistically significant. (I) Schematic showing the experimental strategy. (J) Wholemount fluorescence images of the pancreas from *Ins2-DreER;Ngn3-2A-CreER;NR1* mice after 2 weeks (left) and 12 weeks (right) of Tam treatment. (K) Immunostaining for ZsGreen, tdT, and CK19 or Ins or Sst on pancreatic sections of *Ins2-DreER;Ngn3-2A-CreER;NR1* mice after 2 weeks (top) and 12 weeks (bottom) of Tam treatment. (L, M) Immunostaining for ZsGreen, tdT, and Sst (L) or Ins (M) on pancreatic sections of *Ins2-DreER;Ngn3-2A-CreER;NR1* mice after 2 weeks (top) and 12 weeks (bottom) of Tam treatment. Arrowheads, ZsGreen+Sst+ pancreatic delta cells (L) and tdT+Ins+ pancreatic beta cells (M). (N) Quantification of the percentages of ZsGreen+ or tdT+ cells in Ins+ pancreatic beta cells of *Ins2-DreER;Ngn3-2A-CreER;NR1* mice. Data are mean ± SD; +2 weeks, n = 4 biological replicates; +12 weeks, n = 4 biological replicates; tdT+: $P = 0.09$; n.s., non-significant. In each sample, islets from 10 pancreas sections were quantified. Two-tailed unpaired Student's *t* tests were used for statistical comparisons and $p < 0.05$ was accepted as statistically significant. (O) Cartoon image showing tracing of Ngn3+Ins− non-beta cells and Ins+ beta cells in the pancreas with distinct reporters using *Ins2-DreER;Ngn3-CreER;NR1* and *Ins2-DreER;Ngn3-2A-CreER;R26-NR1* mice. Ngn3+Ins− non-beta cells do not convert to Ins+ beta cells in homeostatic condition. Scale bars, 1 mm (yellow) and 100 μm (white). Each image is representative of 4–5 individual mouse samples. See also Fig. EV3. Source data are available online for this figure.

(Fig. EV3A–F). Next, we treated *Ins2-DreER;Ngn3-CreER;NR1* with Tam to simultaneously trace Ngn3+Ins− non-beta cells (ZsGreen+) and beta cells (tdT+) in the pancreas (Fig. 3A,B). Wholemount imaging revealed sporadic tdT+ islet signals in the pancreas collected 2 weeks after Tam treatment (Fig. 3C). Immunostaining for ZsGreen, tdT, CK19, Ins, and Sst showed that the Ins+ cells in the ductal epithelium were labeled with tdT but not ZsGreen, whereas some ductal Sst+ cells were labeled with ZsGreen (Fig. 3D). In islets, many Sst+ delta cells expressed the ZsGreen reporter but Ins+ beta cells did not express ZsGreen (0.00 ± 0.00%) (Fig. 3E,F). In pancreatic sections collected 12 weeks after Tam, 99.28 ± 0.53% of Ins+ beta cells expressed tdT. Importantly no ZsGreen+tdT−Ins+ beta cells (0.00 ± 0.00%) were observed in ducts or in islets (Fig. 3G,H).

Next, we used *Ngn3-2A-CreER* mice and generated *Ins2-DreER;Ngn3-2A-CreER;NR1* triple knock-in mice (Fig. 3I). Analysis of pancreases collected at 2 weeks and 12 weeks post-Tam revealed that no ZsGreen+ tdT−Ins+ beta cells were generated during the chase period (Fig. 3J–N, 2 weeks: 0.00 ± 0.00%; 12 weeks: 0.00 ± 0.00%). As controls, we did not detect any fluorescent signal in the islets of both *Ins2-DreER;Ngn3-CreER;NR1* and *Ins2-DreER;Ngn3-2A-CreER;NR1* mice in the absence of Tam (Fig. EV3G–N). In summary, these data indicate that during homeostasis, both Ngn3+ ductal cells and Ngn3+ non-beta islet cells (most being Sst+ cells) did not undergo lineage (trans) differentiation toward beta cells (Fig. 3O).

## Ngn3+ non-beta cells gave rise to beta cells after extreme beta-cell loss

Previous studies have reported that non-beta islet cells, including alpha and delta cells, can generate new beta cells after extreme beta cell loss (Chera et al, 2014; Perez-Frances et al, 2021; Thorel et al, 2010; Zhao et al, 2021a). To test whether Ngn3+ non-beta cells could convert to beta cells after extreme beta-cell loss and to provide a positive technical control for our dual-genetic lineage tracing system, we utilized the *R26-R-tdT-DTR* mouse line (Wang et al, 2022) for genetic ablation of beta cells. Upon crossbreeding with *Ins2-DreER*, Dre/rox-mediated expression of diphtheria toxin receptor (DTR) in beta cells can lead to depletion of pre-existing

beta cells after diphtheria toxin (DT) injection (Fig. 4A). We first examined the recombination efficiency of *Ins2-DreER;R26-R-tdT-DTR* by collecting the pancreases 2 weeks after Tam, with multiple DT injections before tissue harvest (Fig. 4B). Wholemount fluorescence revealed tdT signals in PBS-treated but not DT-treated pancreases (Fig. 4C). Consistently, all beta cells in the control group were labeled with tdT whereas virtually all beta cells in the DT-treated group were ablated (Fig. 4D,E), demonstrating high ablation efficiency in this model. Concurrently, there was a notable increase in the population of non-beta endocrine cells, including alpha, delta, and PP cells (Fig. EV4A,B).

Next, we crossed *R26-R-tdT-DTR* with *Ins2-DreER;Ngn3-2A-CreER;NR1* (Fig. 4F). By design, pre-existing beta cells will be ablated after DT treatment, and if Ngn3+ non-beta cells (ZsGreen+) could convert to beta cells, the newly generated beta cells would be ZsGreen+. After Tam treatment and a washout period, we induced DT injection and collected pancreases from *Ins2-DreER;Ngn3-2A-CreER;NR1;R26-R-tdT-DTR* mice 2 weeks or 4 weeks post-Tam (Fig. 4G). At 2 weeks post-Tam, wholemount imaging showed tdT+ islets in the PBS-control group but not in the DT-treated group (Fig. 4H), which was confirmed by sectional analysis (Fig. 4I,J). At 4 weeks post-Tam, several Ins+ beta cells reappeared and 53.92 ± 9.41% of these newly generated beta cells were ZsGreen+ (Fig. 4K,L). Furthermore, these ZsGreen+ cells also expressed Pdx1 and Nkx6.1 (Fig. 4M). Meanwhile, we could hardly find Ins+ cells in the pancreatic duct at this time point. These data suggested that Ngn3+ non-beta cells could convert to beta cells after extreme beta-cell loss, providing a positive technical control for detecting beta-cell neogenesis by our dual-genetic lineage tracing system (Fig. 4M,N).

These Ngn3+ non-beta cells could likely be alpha cells, delta cells or PP cells that convert to beta cells under special conditions (Chera et al, 2014; Perez-Frances et al, 2021; Thorel et al, 2010; Zhao et al, 2021a). Our results showed that with DT treatment, there were 3.20 ± 1.41 zsGreen+ ductal cells per section, compared to 3.60 ± 1.23 zsGreen+ ductal cells per section in the PBS-treated control group (Fig. EV4C–E). Notably, we did not observe a significant increase in the number of Ngn3+ non-beta cells within the ducts following DT treatment (Fig. EV4E). These results strongly suggest that rare Ngn3+ non-beta cells within the ducts are

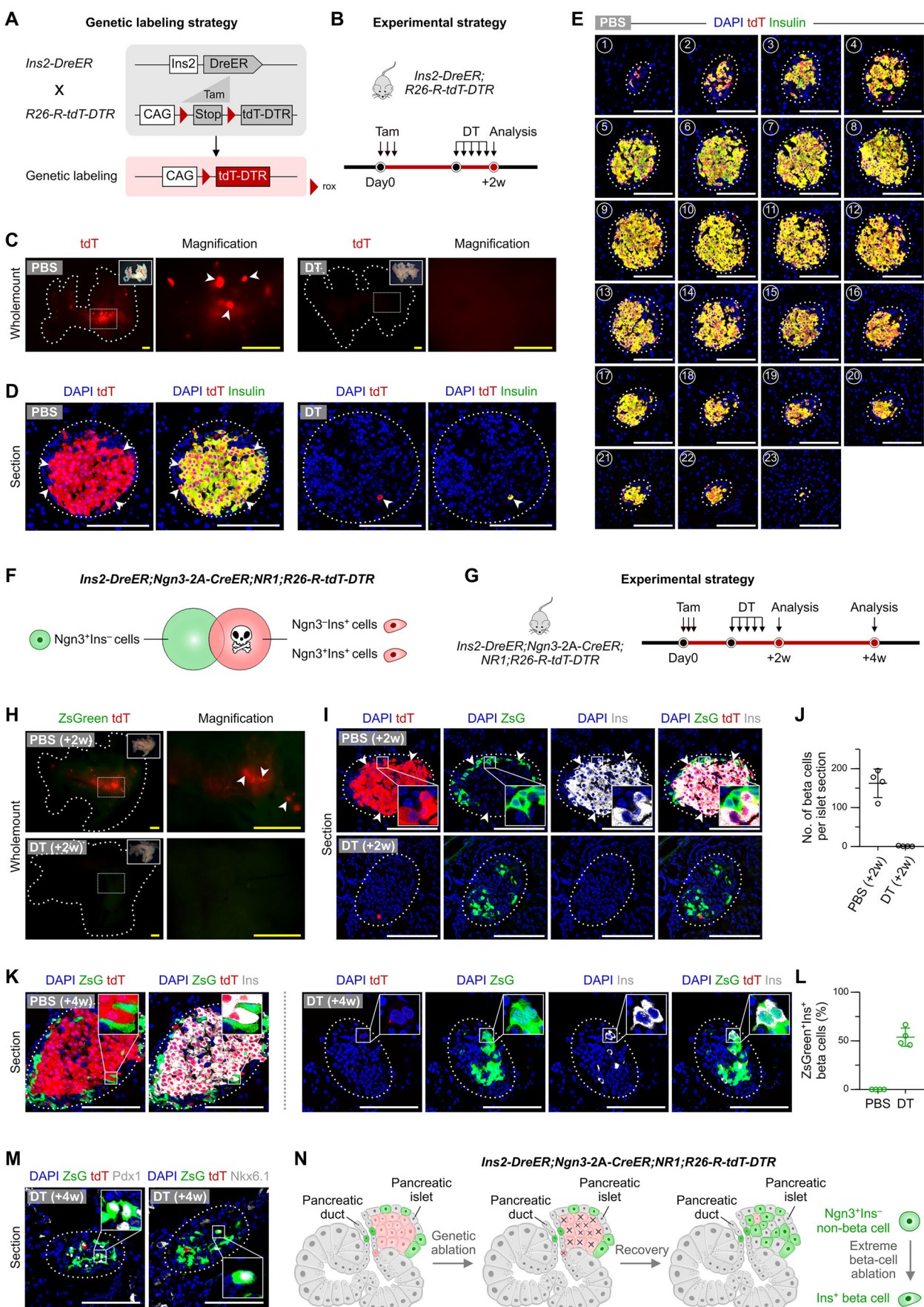

**Figure 4. Ngn3+Ins− non-beta cells contribute to new Ins+ beta cells after extreme loss of beta cells.**

(A) Schematic illustrating the labeling strategy by *Ins2-DreER;R26-R-tdT-DTR* mice. (B) Schematic showing the experimental strategy. (C) Wholemount fluorescence images of the pancreas from *Ins2-DreER;R26-R-tdT-DTR* mice after PBS (left) or DT (right) treatment. Arrowheads, pancreatic islets. (D) Immunostaining for tdT and Ins on pancreatic sections of *Ins2-DreER;R26-R-tdT-DTR* mice after PBS (left) or DT (right) treatment. Arrowheads, tdT+Ins+ pancreatic beta cells. (E) Immunostaining for tdT and Ins on serial sections of the pancreatic islets from *Ins2-DreER;R26-R-tdT-DTR* mice after PBS treatment. (F) Cartoon image showing the labeling strategy by *Ins2-DreER;Ngn3-2A-CreER;NR1;R26-R-tdT-DTR* mice. (G) Schematic showing the experimental strategy. (H) Wholemount fluorescence images of the pancreas from *Ins2-DreER;Ngn3-2A-CreER;NR1;R26-R-tdT-DTR* mice after PBS (top) or DT (bottom) treatment. Arrowheads, pancreatic islets. (I) Immunostaining for ZsGreen, tdT, and Ins on pancreatic sections of *Ins2-DreER;Ngn3-2A-CreER;NR1;R26-R-tdT-DTR* after PBS (top) or DT (bottom) treatment. Arrowheads, tdT+Ins+ pancreatic beta cells. (J) Quantification of the beta cell numbers per islet sections of *Ins2-DreER;Ngn3-2A-CreER;NR1;R26-R-tdT-DTR* mice after PBS or DT treatment. Data are mean ± SD; PBS, $n = 4$ biological replicates; DT, $n = 4$ biological replicates. In each sample, islets from 10 pancreas sections were quantified. (K) Immunostaining for ZsGreen, tdT, and Ins on pancreatic sections of *Ins2-DreER;Ngn3-2A-CreER;NR1;R26-R-tdT-DTR* mice with PBS (left) or DT (right) treatment 4 weeks after Tam treatment. (L) Quantification of the percentages of ZsGreen+ cells in Ins+ pancreatic beta cells of *Ins2-DreER;Ngn3-2A-CreER;NR1;R26-R-tdT-DTR* mice with PBS or DT treatment. Data are mean ± SD; PBS, $n = 4$ biological replicates; DT, $n = 4$ biological replicates. In each sample, islets from 10 pancreas sections were quantified. (M) Immunostaining for ZsGreen, tdT, and Pdx1 and Nkx6.1 on pancreatic sections of *Ins2-DreER;Ngn3-2A-CreER;NR1;R26-R-tdT-DTR* mice with DT treatment 4 weeks after Tam treatment. (N) Cartoon image showing that Ngn3+Ins− non-beta cells contribute to new Ins+ beta cells after genetic ablation of beta cells. Scale bars, 1 mm (yellow) and 100 μm (white). Each image is representative of 4–5 individual mouse samples. See also Fig. EV4. Source data are available online for this figure.

highly unlikely to migrate and convert to islet beta cells, even under conditions of severe beta cell loss.

Previous studies have shown conflicting results regarding whether Ngn3+ progenitors contribute to the generation of new beta cells in the pancreatic ductal ligation (PDL) injury model (Van de Casteele et al, 2013; Xiao et al, 2013a; Xiao et al, 2013b; Xu et al, 2008). Compared to the extreme beta cell ablation injury, the PDL injury causes relatively gentler damage to the islets. To address this issue, we used two mouse models, *Ins2-DreER;Ngn3-CreER;NR1* and *Ins2-DreER;Ngn3-2A-CreER;NR1* (Fig. EV4F,K). We subjected these mice to PDL injury and collected samples for analysis 2 weeks later. Hematoxylin and eosin staining revealed that the pancreatic structure in the unligated head region remained largely normal, while there was a significant accumulation of ductal cells in the ligated tail region. Despite these morphological changes, we did not detect any ZsGreen+tdT−Ins+ beta cells in either the pancreatic head or the tail under these conditions (Fig. EV4F–O). These findings suggest that in the context of PDL injury, Ngn3+ non-beta cells do not contribute to the generation of new beta cells, providing new insights into the mechanisms of beta cell neogenesis in response to different types of pancreatic injuries.

## Ductal epithelial cells do not contribute to beta cell neogenesis during adult homeostasis

To improve lineage tracing of pancreatic ducts, we generated a new mouse knockin line *Hnf1b-2A-CreER* that could label ductal epithelial cells with exceptionally high efficiency. We crossed *Hnf1b-2A-CreER* with *Ins2-DreER;R26-TLR* mice (Fig. 5A). In this dual-genetic lineage tracing system, we can simultaneously label and trace Hnf1b+Ins− non-beta cells (ZsGreen−tdT+), Hnf1b+Ins+ beta cells (ZsGreen+tdT+), and Hnf1b−Ins+ beta cells (ZsGreen+tdT−) with distinct fluorescent markers (Fig. 5A).

We treated *Ins2-DreER;Hnf1b-2A-CreER;R26-TLR* adult mice with Tam and analyzed them after 2 and 12 weeks (Fig. 5B). We performed wholemount fluorescence imaging and found that ZsGreen+ islets expressed strong tdT after Tam induction (Fig. 5C). Quantitively, 99.76 ± 0.26% of CK19+ ductal cells were positive for tdT (Fig. 5D,E). We also found that almost all Ins+ cells lining the ductal epithelium were ZsGreen+tdT+ whereas Sst+ cells in the duct were ZsGreen−tdT+ (Fig. 5F). In the pancreatic islets, while vast majority of beta cells (99.94 ± 0.11%) were labeled with ZsGreen, a portion of the beta cells were also labeled with tdT, but we never detected tdT+ZsGreen− beta cells (0.00 ± 0.00%) (Fig. 5G). After the chase period, all beta cells in ducts and islets were either ZsGreen+tdT− or ZsGreen+tdT+, and there was no tdT+ZsGreen− beta cells (0.00 ± 0.00%) (Fig. 5H–J). Moreover, the proportion of ZsGreen+tdT+ beta cells was not significantly different before and after the chase period (8.32 ± 0.73% for 2 weeks and 9.09 ± 1.25% for 12 weeks, Fig. 5J), indicating that the Hnf1b+Ins+ beta cell subpopulation does not harbor a proliferative advantage over Hnf1b−Ins+ beta cells. As a technical control, we rarely detect ZsGreen+ or tdT+ cells without Tam treatment (Fig. EV5A–F). The above data demonstrated that pancreatic ductal cells did not adopt beta cell fate in the 10-week chase period in the adult pancreas (Fig. 5K).

We next crossed *Hnf1b-2A-CreER* with the *Ins2-DreER;NR1* mouse line in which all beta cells were labeled with tdT whereas Hnf1b+ non-beta cells, including ductal epithelial cells, were labeled with ZsGreen (Fig. 6A). We treated *Ins2-DreER;Hnf1b-2A-CreER;NR1* with Tam and collected the pancreases 2 weeks and 12 weeks later (Fig. 6B). Wholemount imaging revealed tdT+ islets (Fig. 6C). Virtually all ductal cells (99.61 ± 0.35%) were labeled with the ZsGreen reporter (Fig. 6D,E). Moreover, we detected a few tdT+ZsGreen− cells in the ductal epithelium that were stained positive for insulin (Fig. 6F). The Sst+ ductal cells were labeled with ZsGreen (Fig. 6F). Examination of pancreatic sections from *Ins2-DreER;Hnf1b-2A-CreER;NR1* mice revealed that none of the ZsGreen+tdT− cells expressed Ins in the islets (0.00 ± 0.00%) (Fig. 6G). After 12 weeks of Tam treatment, we analyzed the pancreases and found no ZsGreen+tdT− beta cells in either ducts or islets (0.00 ± 0.00%) (Fig. 6H–J). As a control, very few fluorescent cells were detected in the *Ins2-DreER;Hnf1b-2A-CreER;NR1* pancreas without Tam treatment (Fig. EV5G–L), suggesting negligible leakage in this system. Taken together, these results indicated that pancreatic ductal cells did not contribute to new beta cells in adult homeostasis (Fig. 6K).

## Discussion

Over the last two decades, the contribution of Ngn3+ progenitors and pancreatic ductal cells to new beta cells in adults has remained

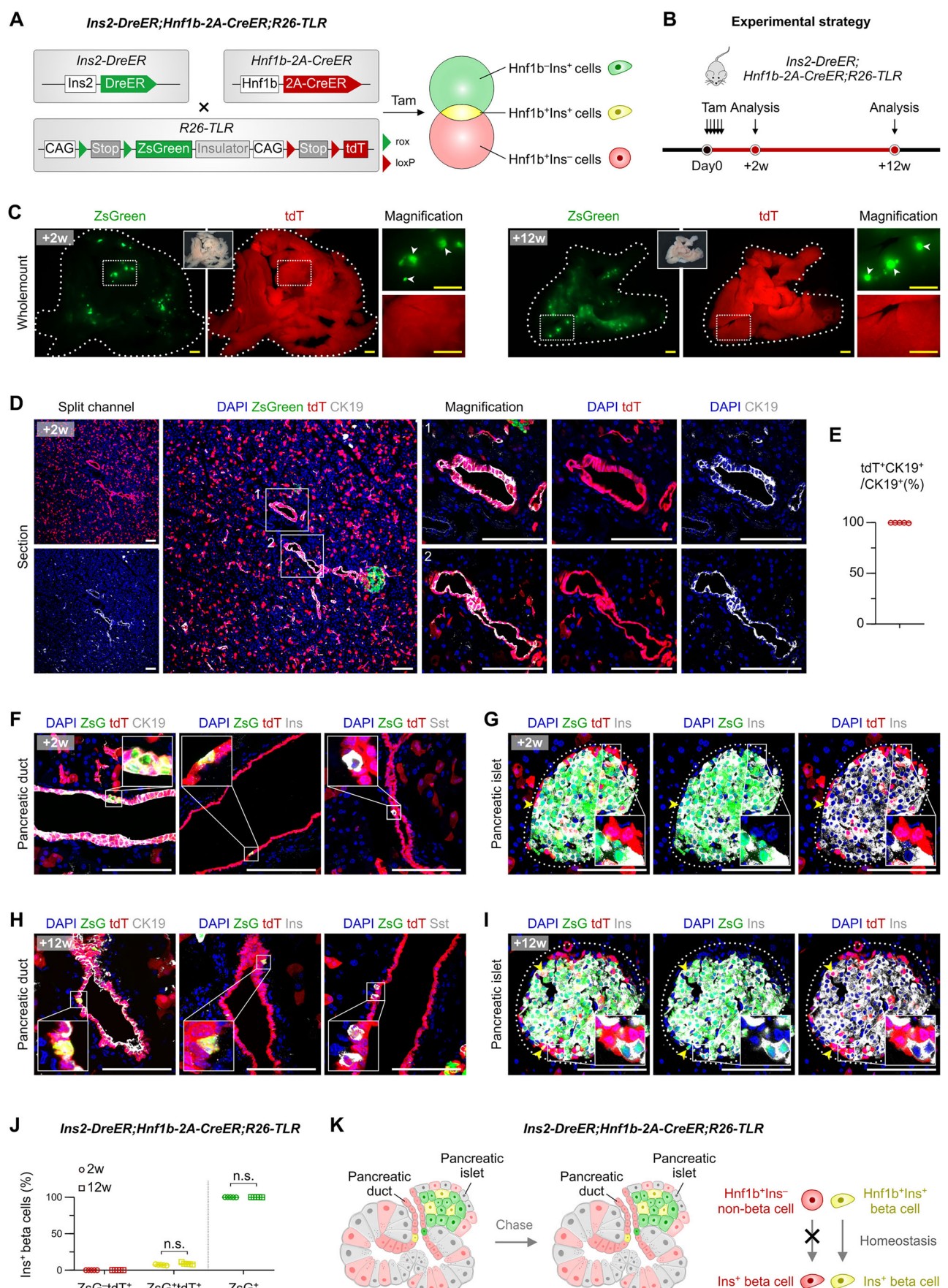

◄

**Figure 5.  Simultaneous tracing of Hnf1b+Ins- non-beta cells, Hnf1b+Ins+ beta cells, and Hnf1b-Ins+ beta cells by the triple light reporter.**

(A) Schematic illustrating the labeling strategy by *Ins2-DreER;Hnf1b-2A-CreER;R26-TLR* mice. (B) Schematic showing the experimental strategy. (C) Wholemount fluorescence images of the pancreas from *Ins2-DreER;Hnf1b-2A-CreER;R26-TLR* 2 weeks (left) and 12 weeks (right) after Tam treatment. Arrowheads, pancreatic islets. (D) Immunostaining for ZsGreen, tdT, and CK19 on pancreatic sections of *Ins2-DreER;Hnf1b-2A-CreER;R26-TLR* mice 2 weeks after Tam treatment. (E) Quantification of the percentage of tdT+CK19+ cells among CK19+ pancreatic ductal cells of *Ins2-DreER;Hnf1b-2A-CreER;R26-TLR* mice. Data are mean ± SD; $n = 5$ biological replicates. In each sample, islets from 10 pancreas sections were quantified. (F) Immunostaining for ZsGreen, tdT, and CK19 or Ins or Sst on pancreatic sections of *Ins2-DreER;Hnf1b-2A-CreER;R26-TLR* mice 2 weeks after Tam treatment (G) Immunostaining for ZsGreen, tdT, and Ins on pancreatic sections of *Ins2-DreER;Hnf1b-2A-CreER;R26-TLR* mice 2 weeks after Tam treatment. Arrowheads, ZsGreen+tdT+Ins+ pancreatic beta cells. (H) Immunostaining for ZsGreen, tdT, and CK19 or Ins or Sst on pancreatic sections of *Ins2-DreER;Hnf1b-2A-CreER;R26-TLR* mice 12 weeks after Tam treatment. (I) Immunostaining for ZsGreen, tdT, and Ins on pancreatic sections of *Ins2-DreER;Hnf1b-2A-CreER;R26-TLR* mice 12 weeks after Tam treatment. Arrowheads, ZsGreen+tdT+Ins+ pancreatic beta cells. (J) Quantification of the percentages of ZsGreen-tdT+ and ZsGreen+tdT+ or ZsGreen+ cells in Ins+ pancreatic beta cells of *Ins2-DreER;Hnf1b-2A-CreER;R26-TLR* mice. Data are mean ± SD; +2 weeks, $n = 5$ biological replicates; +12 weeks, $n = 5$ biological replicates; ZsGreen+tdT+: $P = 0.27$; ZsGreen+: $P = 0.57$; n.s., non-significant. In each sample, islets from 10 pancreas sections were quantified. Two-tailed unpaired Student's *t* tests were used for statistical comparisons and $P < 0.05$ was accepted as statistically significant. (K) Cartoon image showing that *Ins2-DreER;Hnf1b-2A-CreER;R26-TLR* label Hnf1b+Ins- non-beta cells, Hnf1b+Ins+ beta cells, and Hnf1b-Ins+ beta cells in pancreas simultaneously. Hnf1b+Ins- non-beta cells do not contribute to beta cells was detected during homeostasis. Scale bars, 1 mm (yellow) and 100 μm (white). Each image is representative of five individual mouse samples. See also Fig. EV5. Source data are available online for this figure.

highly controversial and requires further investigation to clarify this issue. In this study, we employed a newly developed dual-genetic tracing strategy to address the question of whether Ngn3+ progenitors or other potential progenitors existing in the pancreatic ductal epithelium could contribute to beta cell neogenesis during murine adult homeostasis. Using independent lineage tracing strategies, we provided direct in vivo genetic evidence that Ngn3+ non-beta cells or Hnf1b+ non-beta cells (including virtually all ductal cells) do not contribute to de novo beta cells during adult homeostasis. Our study also demonstrated that Ngn3+ progenitors could generate new beta cells under the condition of extreme ablation of pre-existing beta cells. This transformation serves as a positive technical control for the detection of beta cell neogenesis by our dual-genetic lineage tracing system.

Precise genetic lineage tracing study is essential for elucidating specific cell behaviors and fate conversions (He et al, 2017; Kretzschmar and Watt, 2012; Weng et al, 2022). Previous lineage tracing studies have been performed using conventional Cre/loxP recombination system, with many controversies for either supporting or opposing the adult stem cells in tissue repair and regeneration (He et al, 2017; Weng et al, 2022). However, the Cre recombinase expression in non-specific cells may lead to misinterpretation of lineage tracing data. For example, supportive evidence for cKit+ cardiac stem cells should be based on the fact that *Kit* is not expressed in pre-existing cardiomyocytes, but subsequent studies have shown that Kit is expressed in cardiomyocytes, and unintentional labeling of *Kit-CreER* confounds this interpretation (He et al, 2019, 2017). Pancreatic beta cells are one of the most highly differentiated cells in adults and one of the most important features of the putative beta-cell progenitors is that they do not express insulin themselves. Despite differences in mouse lines, the expression of *Ngn3* and *Hnf1b* in pre-existing beta cells is widely recognized (Magenheim et al, 2023; Solar et al, 2009; Wang et al, 2009), and single-cell-RNA sequencing analyses have also revealed the presence of Ngn3+Ins+ beta cell populations in human beta cells (Li et al, 2020). This is likely to be one of the factors that have given rise to the debate as to whether Ngn3+ progenitors (insulin negative) and ductal cells contribute to beta-cell neogenesis.

Here, we generated new knock-in mouse lines *Ngn3-2A-CreER* and *Hnf1b-2A-CreER*, which are efficient for labeling and do not affect endogenous gene expression compared to previous lines. We found that *Ngn3-2A-CreER* and *Hnf1b-2A-CreER* also label a fraction of beta cells at the pulse time point. Taking advantage of dual-genetic tracing technology, we could distinguish the "ectopic" labeling introduced by Cre recombinase expression in pre-existing beta cells. Using intersectional and nested reporter designs, respectively, we demonstrated that Dre/rox-mediated recombination effectively blocked the "unwanted" Cre reporter expression in beta cells, thereby facilitating the clear interpretation of lineage tracing data. We addressed the long-standing debate on the contribution of Ngn3+ progenitors and pancreatic ductal cells to the beta cells in the adult pancreas. Our dual-genetic lineage tracing data demonstrated that Ngn3+ non-beta cells and pancreatic ductal cells do not generate new beta cells de novo during homeostasis in murine adults.

The dual-genetic lineage tracing strategy not only controls Cre/loxP recombination in "unwanted" cells more precisely but also sensitively captures lineage conversions in vivo. By using the genetic ablation approach, we could also detect the in vivo conversion of Ngn3+ non-beta cells to beta cells under certain conditions. However, a limitation of our approach is that it is unable to distinguish between Ngn3+ islet cells and Ngn3+ ductal cells as the exact source for newly generated insulin-expressing cells. This is because *Ngn3-2A-CreER* labels subpopulations of alpha, delta, and PP cells in the islets, as well as Ngn3-expressing cells located in the ducts, all of which are labeled by zsGreen. Based on previous studies, we postulated that these Ngn3+ non-beta cells responsible for the generation of Ins+ cells are predominantly subpopulations of endocrine cells within the pancreatic islets (Chera et al, 2014; Perez-Frances et al, 2021; Thorel et al, 2010). Our model provides a rigorous technical standard and an unprecedented strategy for revisiting the contribution of putative pancreatic stem cells or progenitors in beta cell renewal and regeneration. Our study does not examine the potential of Ngn3+ ductal progenitor cells for beta-cell neogenesis under various injury models or some extreme conditions. However, our study highlights the urgent need for further research in this debated area. Our findings convincingly demonstrate that during homeostasis, Ngn3+ non-beta cells, which include the potential Ngn3+ ductal or islet progenitors, do not contribute to beta-cell neogenesis in the adult pancreas. Currently, it remains unclear which type of injury

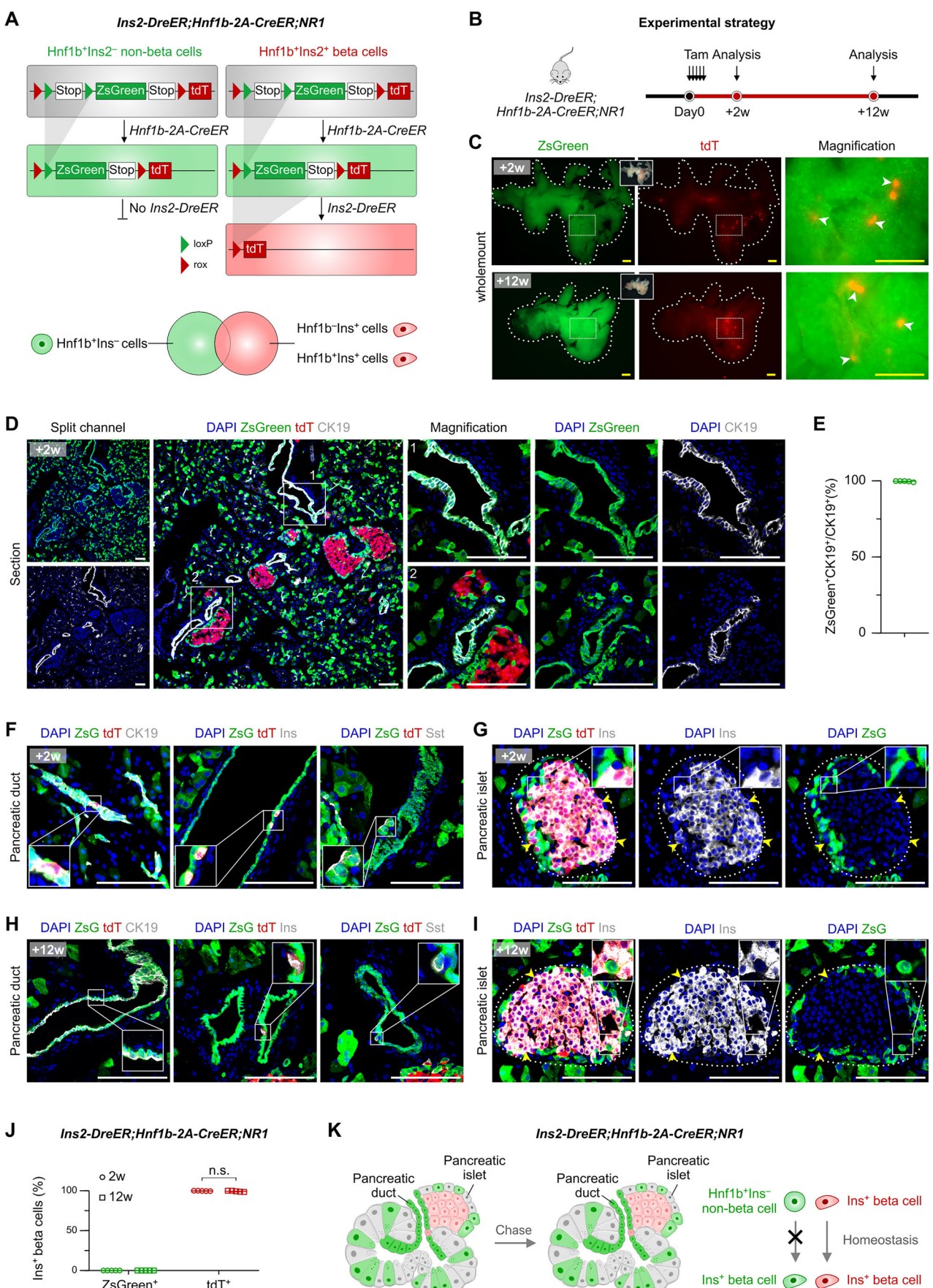

**Figure 6.   Nested strategy shows Hnf1b$^+$Ins$^-$ non-beta cells do not contribute to Ins$^+$ beta cells during adult homeostasis.**

(A) Schematic of the labeling strategy by *Ins2-DreER;Hnf1b-2A-CreER;NR1* mice. (B) Schematic showing the experimental strategy. (C) Wholemount fluorescence images of the pancreas from *Ins2-DreER;Hnf1b-2A-CreER;NR1* 2 weeks and 12 weeks after Tam treatment. Arrowheads, pancreatic islets. (D) Immunostaining for ZsGreen, tdT, and CK19 on pancreatic sections of *Ins2-DreER;Hnf1b-2A-CreER;NR1* mice 2 weeks after Tam treatment. (E) Quantification of the percentage of ZsGreen$^+$CK19$^+$ cells in CK19$^+$ pancreatic ductal cells of *Ins2-DreER;Hnf1b-2A-CreER;NR1* mice. Data are mean ± SD; $n = 5$ biological replicates. In each sample, islets from ten pancreas sections were quantified. (F) Immunostaining for ZsGreen, tdT, and CK19 or Ins or Sst on pancreatic sections of *Ins2-DreER;Hnf1b-2A-CreER;NR1* mice 2 weeks after Tam treatment. (G) Immunostaining for ZsGreen, tdT, and Ins on pancreatic sections of *Ins2-DreER;Hnf1b-2A-CreER;NR1* mice 2 weeks after Tam treatment. Arrowheads, tdT$^+$Ins$^+$ pancreatic beta cells. (H) Immunostaining for ZsGreen, tdT, and CK19 or Ins or Sst on pancreatic sections of *Ins2-DreER;Hnf1b-2A-CreER;NR1* mice 12 weeks after Tam treatment. (I) Immunostaining for ZsGreen, tdT, and Ins on pancreatic sections of *Ins2-DreER;Hnf1b-2A-CreER;NR1* mice 12 weeks after Tam treatment. Arrowheads, tdT$^+$Ins$^+$ pancreatic beta cells. (J) Quantification of the percentage of ZsGreen$^+$ or tdT$^+$ cells in Ins$^+$ pancreatic beta cells of *Ins2-DreER;Hnf1b-2A-CreER;NR1* mice. Data are mean ± SD; +2 weeks, $n = 5$ biological replicates; +12 weeks, $n = 5$ biological replicates; tdT$^+$: $P = 0.21$; n.s. non-significant. In each sample, islets from 10 pancreas sections were quantified. Two-tailed unpaired Student's $t$ tests were used for statistical comparisons and $P < 0.05$ was accepted as statistically significant. (K) Cartoon image showing that *Ins2-DreER;Hnf1b-2A-CreER;NR1* label Hnf1b$^+$Ins$^-$ non-beta cells and Ins$^+$ beta cells in pancreas simultaneously. Hnf1b$^+$Ins$^-$ non-beta cells do not generate new beta cells in adult homeostasis. Scale bars, 1 mm (yellow) and 100 μm (white). Each image is representative of five individual mouse samples. See also Fig. EV5. Source data are available online for this figure.

or extreme condition may unveil the regenerative capacity of rare Ngn3$^+$ duct cells to contribute to beta-cell regeneration. In the case of the *Hnf1b-2A-CreER* line, although it labeled ductal cells with high-efficiency, it also labeled a subset of islet endocrine cells. Therefore, using the dual-genetic tracing strategy based on the intersection of Ngn3 and Hnf1b (for example, *Ngn3-2A-CreER;Hnf1b-2A-DreER;R26-RSR-LSL-tdT*) cannot specifically target the potential Ngn3$^+$ ductal progenitors. There is no available genetic tool that can definitively resolve whether rare Ngn3$^+$ duct cells contribute to beta cell neogenesis after severe beta cell loss. However, we did not observe a significant increase in the number of Ngn3$^+$ non-beta cells within the ducts following DT treatment. These results strongly suggest that these rare Ngn3$^+$ non-beta cells within the ducts are highly unlikely to migrate and convert to islet beta cells, even under conditions of severe beta cell loss. This finding provides quantitative support that these Ngn3$^+$ non-beta cells within the ducts are unlikely to play a meaningful biological role in islet beta cell neogenesis. Future studies should also focus on the as-yet-unrevealed mechanism driving beta cell regeneration in the adult pancreas, including the promotion of replication of pre-existing beta cells or forced expression of exogenous reprogramming factors for beta cell regeneration.

## Methods

### Reagents and Tools Table

| Reagent/resource | Reference or source | Identifier or catalog number |
| --- | --- | --- |
| **Experimental models** | | |
| *Ngn3-CreER* | Gu et al, 2002 | N/A |
| *R26-tdT* | Madisen et al, 2010 | N/A |
| *Ins2-DreER* | Zhao et al, 2021a | N/A |
| *R26-TLR* | Liu et al, 2020 | N/A |
| *NR1* | He et al, 2017 | N/A |
| *R26-R-tdT-DTR* | Wang et al, 2022 | N/A |
| *Ngn3-2A-CreER* | This study | N/A |
| *Hnf1b-2A-CreER* | This study | N/A |
| **Recombinant DNA** | | |
| N/A | | |

| Reagent/resource | Reference or source | Identifier or catalog number |
| --- | --- | --- |
| **Antibodies** | | |
| Goat anti-tdTomato | Rockland | Cat# 200-101-379; RRID: AB_2744552 |
| Rabbit anti-tdTomato | Rockland | Cat# 600-401-379; RRID: AB_2209751 |
| Rat anti-tdTomato | Proteintech (ChromoTek) | Cat# 5F8; RRID: AB_2336064 |
| Rabbit anti-ZsGreen | Clontech | Cat# 632474; RRID: AB_2491179 |
| Guinea pig-Insulin | Dako Agilent | Cat# A0564; RRID: AB_10013624 |
| Rabbit anti-Insulin | Abcam | Cat# ab63820; RRID: AB_1925116 |
| Rat anti-CK19 | Developmental Studies Hybridoma Bank | Cat# TROMA-III; RRID: AB_2133570 |
| Goat anti-E-cad | R & D systems | Cat# AF748; RRID: AB_355568 |
| Rabbit anti-Glucagon | Sigma-Aldrich | Cat# SAB4501137; RRID: AB_10761583 |
| Rat anti-Somatostatin | Santa Cruz | Cat# sc-47706; RRID: AB_628268 |
| Rabbit anti-Somatostatin | Abcam | Cat# ab111912; RRID: AB_10903864 |
| Goat anti-Pancreatic Polypeptide | Abcam | Cat# ab77192; RRID: AB_1524152 |
| Donkey anti-rabbit 488 | Invitrogen | Cat# A21206; RRID: AB_2535792 |
| Donkey anti-rabbit 555 | Invitrogen | Cat# A31572; RRID: AB_162543 |
| Donkey anti-rabbit 647 | Invitrogen | Cat# A31573; RRID: AB_2536183 |
| Donkey anti-rat 594 | Jackson ImmunoResearch Inc | Cat# 712-585-153; RRID: AB_2340689 |
| Donkey anti-rat 647 | Abcam | Cat# ab150155; RRID: AB_2813835 |
| Donkey anti-goat 555 | Invitrogen | Cat# A21432; RRID: AB_2535853 |
| Donkey anti-goat 647 | Invitrogen | Cat# A21447; RRID: AB_141844 |
| JIR Cy5 Donkey a-Guinea Pig IgG(H + L) | Jackson ImmunoResearch Inc | Cat# 706-175-148; RRID: AB_2340462 |
| PCR primers - *Ngn3-2A-CreER* genotyping - mut | This study | 5'-TACTCCCCAGTCTCCCAAGC-3'; 5'-CGCGCGCCTGAAGATATAGA-3' |

| Reagent/resource | Reference or source | Identifier or catalog number |
|---|---|---|
| PCR primers - Ngn3-2A-CreER genotyping - wt | This study | 5'-ACAAAGATCGAGACCCTGCG-3'; 5'-CCTGTGAGAGGGCTAGGGAT-3' |
| PCR primers - Hnf1b-2A-CreER genotyping - mut | This study | 5'-CCTGGCCATTCCATCCAAGT-3'; 5'-GTTGCATCGACCGGTAATGC-3' |
| PCR primers - Hnf1b-2A-CreER genotyping - wt | This study | 5'-AAGTGCCCACCGTATTCCAG-3'; 5'-ACTTGGCATCTTGGGAGAGC-3' |
| **Chemicals, enzymes and other reagents** | | |
| Tamoxifen | Sigma-Aldrich | Cat# T5648 |
| Diphtheria toxin | Sigma-Aldrich | Cat# D0564 |
| Paraformaldehyde (PFA) | Sigma-Aldrich | Cat# P6148 |
| O.C.T. | Sakura | Cat# 4583 |
| Normal donkey serum | Jackson ImmunoResearch Inc | Cat# 017-000-121 |
| Triton X-100 | Sigma-Aldrich | Cat# X100 |
| PBS | Invitrogen | Cat# C10010500BT |
| DAPI | Invitrogen | Cat# D21490 |
| UltraPure Distilled Water | Invitrogen | Cat# 10977023 |
| **Software** | | |
| Fiji | https://ImageJ.net/software/fiji/ | N/A |
| GraphPad Prism 8 software | https://www.graphpad.com | N/A |
| PhotoLine | https://www.pl32.com/ | N/A |
| **Other** | | |
| N/A | | |

## Mice

All animal experiments were performed under the guidelines of the Institutional Animal Care and Use Committee of the State Key Laboratory of Cell Biology, Center for Excellence in Molecular Cell Science, University of Chinese Academy of Science, Chinese Academy of Science. All mice used in this study were maintained on a C57BL6/ICR mixed background. Both male and female mice were included in the study, and the starting time point (Day0) involved adult mice aged 8–10 weeks. The Ngn3-CreER (Gu et al, 2002), R26-tdT (Madisen et al, 2010), Ins2-DreER (Zhao et al, 2021a), R26-TLR (Liu et al, 2020), NR1 (He et al, 2017), and R26-R-tdT-DTR (Wang et al, 2022) mouse lines have been reported previously. The Ngn3-2A-CreER mouse line was generated by inserting the 2A-CreER cassette after the endogenous Ngn3 gene via homologous recombination. The Hnf1b-2A-CreER mouse line was generated by inserting the 2A-CreER cassette after the endogenous Hnf1b gene via homologous recombination. Both Ngn3-2A-CreER and Hnf1b-2A-CreER mice were generated by the Shanghai Model Organisms Center, Inc. Tam (Sigma, T5648) was dissolved in corn oil and was introduced by gavage (0.2 mg/g) at the indicated time points. DT (Sigma, D0564) was injected intraperitoneally (10 ng/g) at the indicated time points.

## Genomic PCR

Mouse tails or tissues were lysed in lysis buffer consisting of 100 mM Tris-HCl (adjusted to pH 7.8), 0.2% SDS, 5 mM EDTA, 200 mM NaCl, and 100 µg/ml Proteinase K at 55 °C for more than 6 h. After lysis, the mixture was centrifuged at maximum speed for 8 min, and the supernatant was separated and collected. The supernatant was then precipitated with isopropanol and centrifuged again at maximum speed for 5 min to obtain genomic DNA, which was then washed with 70% ethanol and dissolved in deionized water. To determine the genotype of all mice, specific primers adapted to the knock-in sequence were used to distinguish from the wild-type allele.

## Wholemount fluorescence imaging

Wholemount fluorescence imaging was performed as previously described (He et al, 2017). Pancreatic tissues were washed three times in phosphate-buffered saline (PBS) and then fixed in 4% paraformaldehyde (PFA) at 4 °C for 1 h and washed three times with PBS. Then the tissues were placed in PBS, and wholemount brightfield and fluorescence imaging were taken by a Zeiss stereo microscope (Zeiss Axio Zoom. V16).

## Immunostaining

The immunostaining protocol was performed as described previously (He et al, 2017). Briefly, pancreatic tissues were fixed in 4% PFA at 4 °C for 1 h and then washed with PBS. Then the tissues were dehydrated in 30% sucrose at 4 °C overnight and embedded in the optimum cutting temperature (O.C.T., Sakura). 10 µm cryosections of each block were collected on slides. For immunostaining, slides were air-dried at room temperature and washed with PBS. Next, the slides were blocked with 5% PBSST (0.1% Triton X-100 and 5% normal donkey serum in PBS) blocking buffer at room temperature for 30 min and then incubated with primary antibodies at 4 °C overnight. The primary antibodies used were as follows: tdT (Rockland, 200-101-379, 1:1000), tdT (Rockland, 600-401-379, 1:1000), tdT (ChromoTek, 5F8, 1:200), ZsGreen (Clontech, 632474, 1:1000), Insulin (Dako, A0564, 1:500), Insulin (Abcam, ab63820, 1:500), CK19 (Developmental Studies Hybridoma Bank, TROMA-III, 1:500), E-cad (R&D, AF748, 1:500), Glucagon (Sigma, SAB4501137, 1:500), Somatostatin (Santa Cruz, sc-47706, 1:500), Somatostatin (Abcam, ab111912, 1:500), Pancreatic Polypeptide (Abcam, ab77192, 1:200), Pdx1 (Abcam, ab47267, 1:500), and Nkx6.1 (Abcam, ab221549, 1:500). The slides were washed three times with PBS to remove the primary antibodies in the next day, and incubated with Fluor-conjugated secondary antibodies at room temperature for 30 min. After three washes with PBS, the slides were mounted with a mounting medium containing the nuclear stain DAPI. Immunostaining images were acquired with Olympus confocal microscopes (FV1200, FV3000 and FV4000) and a Zeiss confocal microscope (Zeiss LSM880), and analyzed with ImageJ (NIH).

## Pancreatic ductal ligation

Pancreatic ductal ligation (PDL) injury was performed as previously described (Zhao et al, 2021a). In brief, adult mice were first anesthetized in a sealed chamber using 2% isoflurane gas. Once anesthetized, the mice were immediately transferred to a pre-warmed heat pad to ensure the maintenance of their body temperature. Anesthesia was continuously sustained through the inhalation of isoflurane throughout the procedure. Subsequently, the abdominal fur of the mice was carefully removed. A midline incision was then made through the abdominal skin and underlying muscles. The pancreatic head and the duodenum were gently lifted from the retroperitoneum to expose the relevant anatomical structures. The pancreatic duct, located to the left side of the portal vein, was carefully ligated, which demarcates the splenic and gastro-duodenal lobes of the pancreas. After surgery, each mouse was placed in an individual cage and provided with a standard laboratory chow diet and tap water for postoperative recovery.

## H.E. staining

H.E. staining was performed according to a previously described method (Zhao et al, 2021a). First, 10 μm thick cryosections were prepared. These sections were then immersed in PBS for 15 min to wash them. After that, the sections were transferred to hematoxylin A solution and incubated for 3 min. Following the hematoxylin staining, the sections were rinsed three times under running tap water. Subsequently, they were immersed in a solution of 1% concentrated hydrochloric acid diluted in 70% ethanol for 1 min and then washed three times with water. Then the sections were incubated in 1% ammonia water for 1 min and washed three times with water. Next, the sections were stained with Eosin-Y solution for 8–10 sections. After eosin staining, the sections were dehydrated by passing them through a series of ethanol and xylene solutions. Finally, the stained sections were mounted with a resinous medium. All images of the stained sections were captured using an Olympus microscope (Olympus, DP72).

## Quantification

All data were presented as the mean values ± standard deviations (SD). All mice were randomly assigned to different groups. The "$n$" in the figure legends represented the number of biological replicates used in each experiment. The manual counting was used for cell quantification and the raw cell count numbers per section and per individual mouse used in the quantifications were presented in the Appendix Table (Appendix Tables S1–11). Two-tailed unpaired Student's $t$ tests were used for statistical comparisons and $P < 0.05$ was accepted as statistically significant. No statistical methods were used to estimate sample size. No blinding was done. No sample were excluded from the analysis.

## Data availability

This study includes no data deposited in external repositories.

The source data of this paper are collected in the following database record: biostudies:S-SCDT-10_1038-S44318-025-00434-z.

## Peer review information

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

## Acknowledgements

This study was supported by the National Key Research & Development Program of China (2024YFA1803302, 2023YFA1800700, 2023YFA1801300, 2022YFA1104200), the National Natural Science Foundation of China (82088101, 32370802, 32100585, 32370897, 32100648), CAS Project for Young Scientists in Basic Research (YSBR-012), Shanghai Pilot Program for Basic Research-CAS, Shanghai Branch (JCYJ-SHFY-2021-006), Research Funds of Hangzhou Institute for Advanced Study (2022ZZ01015, B04006C01600515), Shanghai Municipal Science and Technology Major Project, CAS-Croucher Funding Scheme for Joint Laboratories, and the New Cornerstone Science Foundation through the New Cornerstone Investigator Program and the XPLORER PRIZE. We would like to thank Yanqing Qin from Institutional Center for Shared Technologies and Facilities of SINH, CAS for technical assistance. We also thank the members of cell platform, and Baojin Wu, Guoyuan Chen, Jinmei Zhang, and Xiaorui Zhang for animal husbandry in Center for Excellence in Molecular Cell Science. We also thank Shanghai Model Organisms Center for mice generation.

## Author contributions

**Xiuzhen Huang**: Data curation; Validation. **Huan Zhao**: Data curation; Supervision; Funding acquisition; Validation; Writing—original draft. **Hui Chen**: Data curation; Validation; Methodology. **Zixin Liu**: Data curation; Formal analysis. **Kuo Liu**: Data curation. **Zan Lv**: Data curation. **Xiuxiu Liu**: Data curation. **Ximeng Han**: Data curation. **Maoying Han**: Data curation. **Jie Lu**: Writing—review and editing. **Qiao Zhou**: Writing—review and editing. **Bin Zhou**: Conceptualization; Resources; Supervision; Funding acquisition; Writing—review and editing.

Source data underlying figure panels in this paper may have individual authorship assigned. Where available, figure panel/source data authorship is listed in the following database record: biostudies:S-SCDT-10_1038-S44318-025-00434-z.

## Disclosure and competing interests statement

The authors declare no competing interests.

# Expanded View Figures

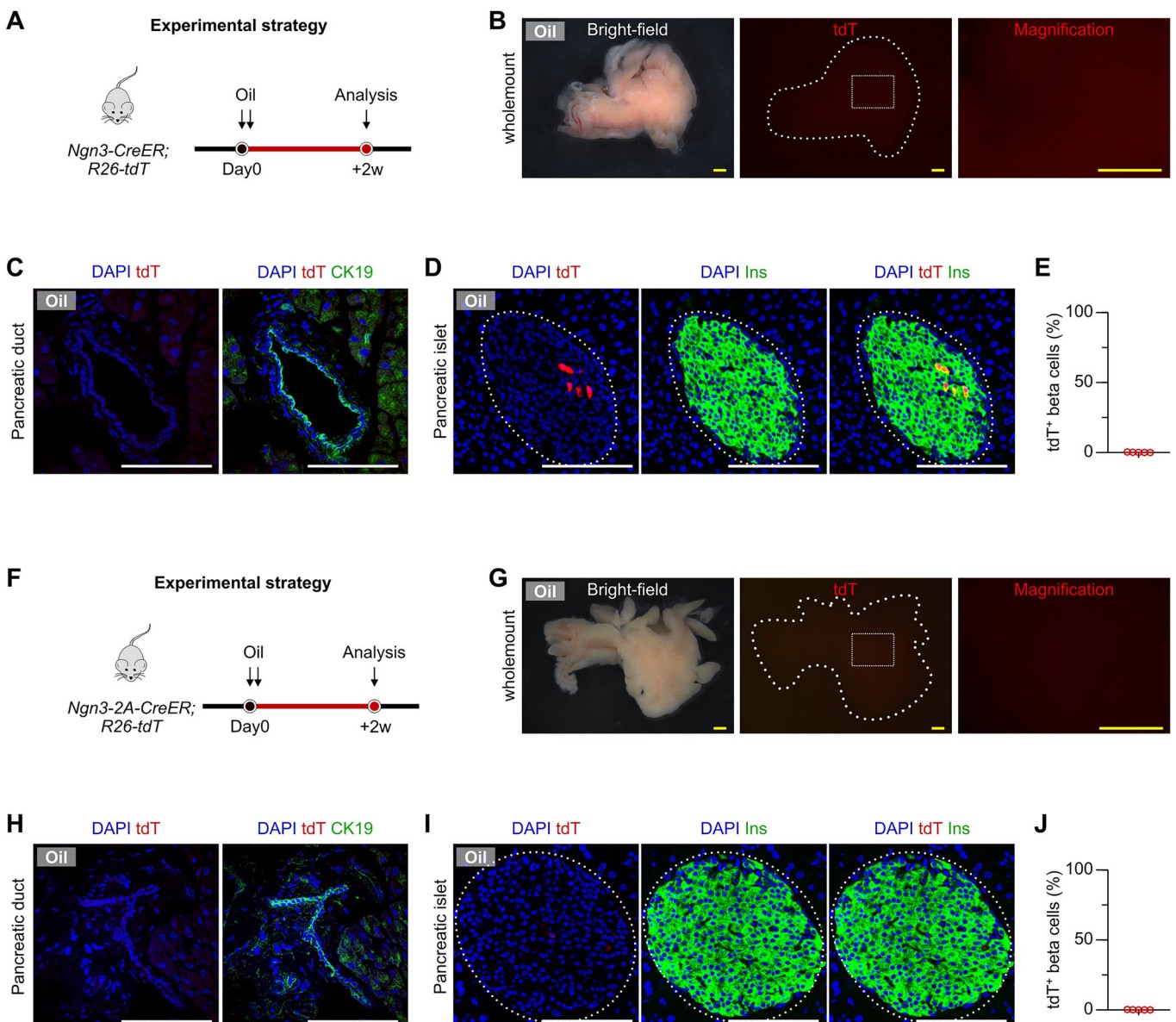

**Figure EV1. Characterization of *Ngn3-CreER;R26-tdT* and *Ngn3-2A-CreER;R26-tdT* without tamoxifen treatment, related to Fig. 1.**

(A) Schematic showing the experimental strategy of *Ngn3-CreER;R26-tdT* with oil treatment. (B) Whole-mount fluorescent images of pancreas collected from *Ngn3-CreER;R26-tdT* after oil treatment. (C, D) Immunostaining for tdT and CK19 (C) or Insulin (Ins, D) on pancreatic sections of *Ngn3-CreER;R26-tdT* after oil treatment. (E) Quantification of the percentage of tdT+ cells in Ins+ beta cells of *Ngn3-CreER;R26-tdT* after oil treatment. Data are mean ± SD; $n = 5$ biological replicates. In each sample, islets from 10 pancreas sections were quantified. (F) Schematic showing the experimental strategy of *Ngn3-2A-CreER;R26-tdT* with oil treatment. (G) Whole-mount fluorescent images of pancreas collected from *Ngn3-2A-CreER;R26-tdT* after oil treatment. (H, I) Immunostaining for tdT and CK19 (H) or Ins (I) on pancreatic sections of *Ngn3-2A-CreER;R26-tdT* after oil treatment. (J) Quantification of the percentage of tdT+ cells in Ins+ beta cells of *Ngn3-2A-CreER;R26-tdT* after oil treatment. Data are mean ± SD; $n = 5$ biological replicates. In each sample, islets from 10 pancreas sections were quantified. Scale bars, yellow, 1 mm; white, 100 μm. Each image is representative of 5 individual samples.

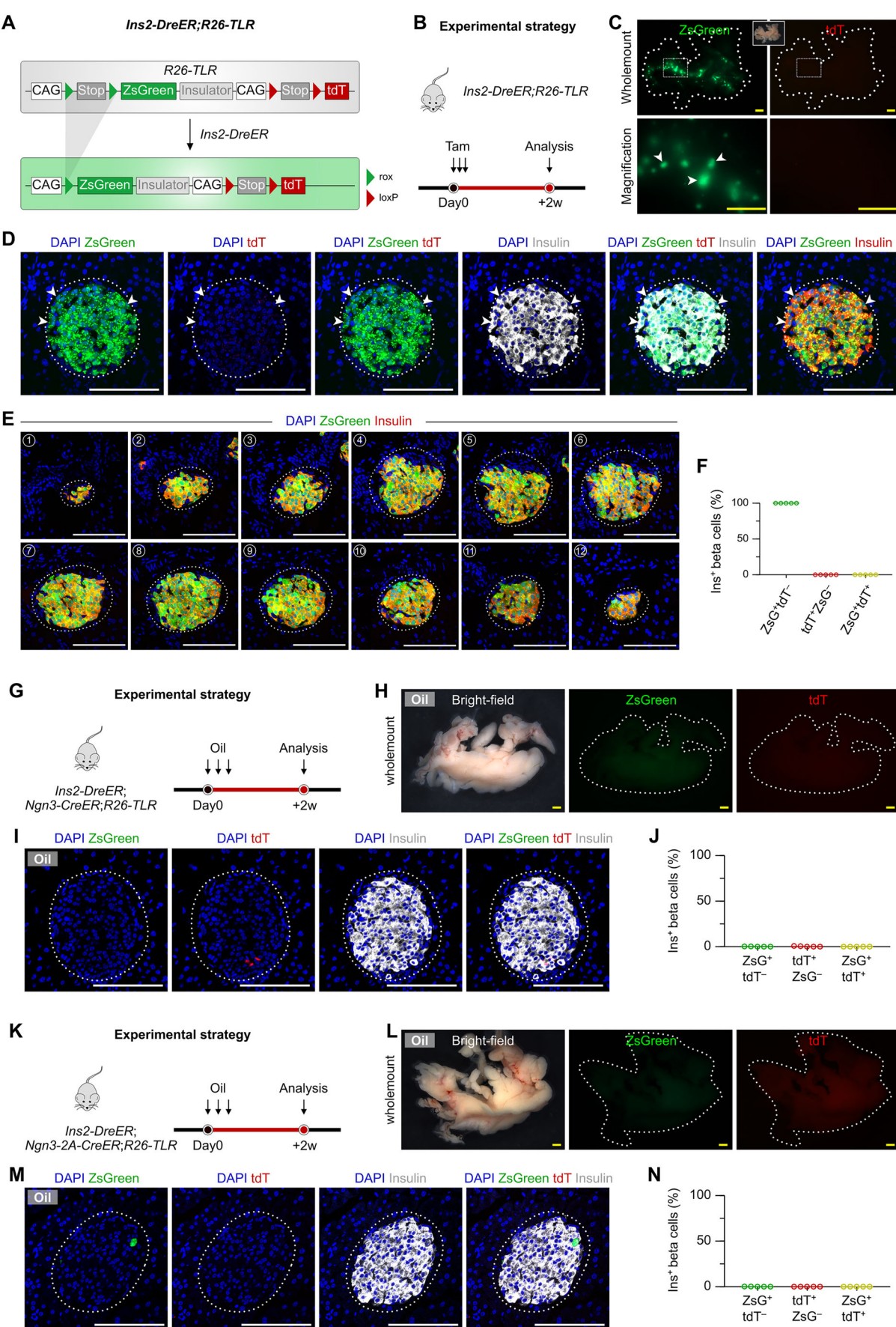

◀ **Figure EV2. Characterization of *Ins2-DreER;R26-TLR*, and characterization of *Ins2-DreER;Ngn3-CreER;R26-TLR* and *Ins2-DreER;Ngn3-2A-CreER;R26-TLR* without tamoxifen treatment, related to Fig. 2.**

(A) Schematic showing the labeling strategy of *Ins2-DreER;R26-TLR*. (B) Schematic showing the experimental strategy of *Ins2-DreER;R26-TLR* with tamoxifen (Tam) treatment. (C) Whole-mount fluorescent images of pancreas from *Ins2-DreER;R26-TLR*. Arrowheads, zsGreen⁺ islets. (D) Immunostaining for ZsGreen, tdT and Ins on pancreatic sections of *Ins2-DreER;R26-TLR* with Tam treatment. Arrowheads, zsGreen⁺ beta cells. (E) Immunostaining for ZsGreen and Ins on serial sections of one pancreatic islet from *Ins2-DreER;R26-TLR* with Tam treatment. (F) Quantification of the percentages of ZsGreen⁺tdT⁻ or tdT⁺ZsGreen⁻ or ZsGreen⁺tdT⁺ cells in Ins⁺ pancreatic beta cells of *Ins2-DreER;R26-TLR*. Data are mean ± SD; $n = 5$ biological replicates. In each sample, islets from 10 pancreas sections were quantified. (G) Schematic showing the experimental strategy of *Ins2-DreER;Ngn3-CreER;R26-TLR* with oil treatment. (H) Whole-mount fluorescent images of pancreas from *Ins2-DreER;Ngn3-CreER;R26-TLR* after oil treatment. (I) Immunostaining for ZsGreen, tdT and Ins on pancreatic sections of *Ins2-DreER;Ngn3-CreER;R26-TLR* after oil treatment. (J) Quantification of the percentage of ZsGreen⁺tdT⁻ or tdT⁺ZsGreen⁻ or ZsGreen⁺tdT⁺ cells in Ins⁺ pancreatic beta cells of *Ins2-DreER;Ngn3-CreER;R26-TLR* after oil treatment. Data are mean ± SD; $n = 5$ biological replicates. In each sample, islets from 10 pancreas sections were quantified. (K) Schematic showing the experimental strategy of *Ins2-DreER;Ngn3-2A-CreER;R26-TLR* with oil treatment. (L) Whole-mount fluorescent images of pancreas from *Ins2-DreER;Ngn3-2A-CreER;R26-TLR* after oil treatment. (M) Immunostaining for ZsGreen, tdT and Ins on pancreatic sections of *Ins2-DreER;Ngn3-2A-CreER;R26-TLR* after oil treatment. (N) Quantification of the percentage of ZsGreen⁺ or tdT⁺ or ZsGreen⁺tdT⁺ cells in Ins⁺ pancreatic beta cells of *Ins2-DreER;Ngn3-2A-CreER;R26-TLR* after oil treatment. Data are mean ± SD; $n = 5$ biological replicates. In each sample, islets from 10 pancreas sections were quantified. Scale bars, yellow, 1 mm; white, 100 μm. Each image is representative of 5 individual samples.

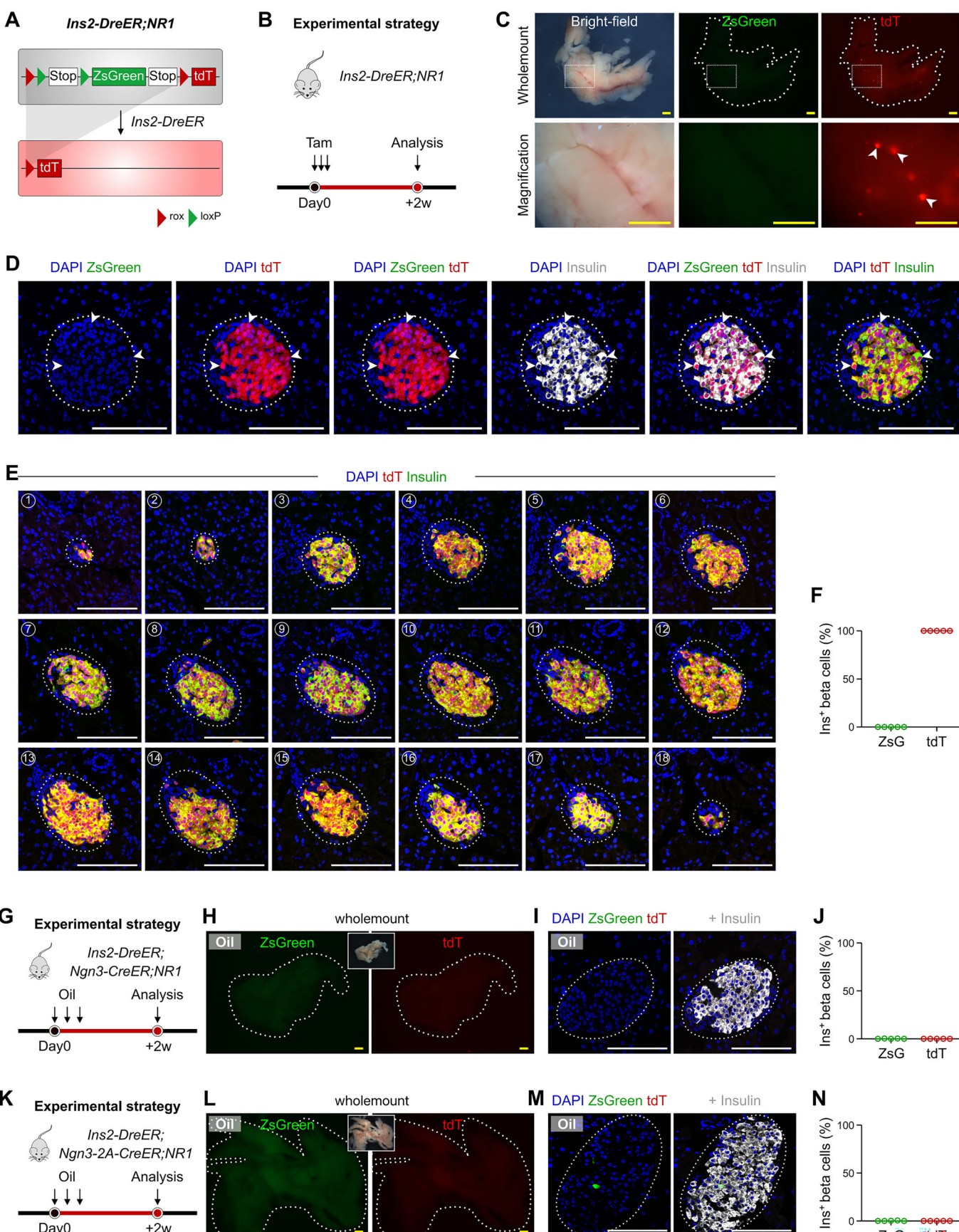

**Figure EV3.  Characterization of *Ins2-DreER;NR1*, and characterization of *Ins2-DreER;Ngn3-CreER;NR1* and *Ins2-DreER;Ngn3-2A-CreER;NR1* without tamoxifen treatment, related to Fig. 3.**

(A) Schematic showing the labeling strategy of *Ins2-DreER;NR1*. (B) Schematic showing the experimental strategy of *Ins2-DreER;NR1* with Tam treatment. (C) Whole-mount fluorescent images of pancreas from *Ins2-DreER;NR1* with Tam treatment. Arrowheads, tdT$^+$ islets. (D) Immunostaining for ZsGreen, tdT and Ins on pancreatic sections of *Ins2-DreER;NR1* with Tam treatment. Arrowheads, tdT$^+$ beta cells. (E) Immunostaining for tdT and Ins on serial sections of one pancreatic islet from *Ins2-DreER;NR1* with Tam treatment. (F) Quantification of the percentage of ZsGreen$^+$ or tdT$^+$ cells in Ins$^+$ pancreatic beta cells of *Ins2-DreER;NR1*. Data are mean ± SD; $n = 5$ biological replicates. In each sample, islets from 10 pancreas sections were quantified. (G) Schematic showing the experimental strategy of *Ins2-DreER;Ngn3-CreER;NR1* with oil treatment. (H) Whole-mount fluorescent images of pancreas from *Ins2-DreER;Ngn3-CreER;NR1* after oil treatment. (I) Immunostaining for ZsGreen, tdT and Ins on pancreatic sections of *Ins2-DreER;Ngn3-CreER;NR1* after oil treatment. (J) Quantification of the percentage of ZsGreen$^+$ or tdT$^+$ cells in Ins$^+$ pancreatic beta cells of *Ins2-DreER;Ngn3-CreER;NR1* after oil treatment. Data are mean ± SD; $n = 5$ biological replicates. In each sample, islets from 10 pancreas sections were quantified. (K) Schematic showing the experimental strategy of *Ins2-DreER;Ngn3-2A-CreER;NR1* with oil treatment. (L) Whole-mount fluorescent images of pancreas from *Ins2-DreER;Ngn3-2A-CreER;NR1* after oil treatment. (M) Immunostaining for ZsGreen, tdT and Ins on pancreatic sections of *Ins2-DreER;Ngn3-2A-CreER;NR1* after oil treatment. (N) Quantification of the percentage of ZsGreen$^+$ or tdT$^+$ cells in Ins$^+$ pancreatic beta cells of *Ins2-DreER;Ngn3-2A-CreER;NR1* after oil treatment. Data are mean ± SD; $n = 5$ biological replicates. In each sample, islets from 10 pancreas sections were quantified. Scale bars, yellow, 1 mm; white, 100 μm. Each image is representative of 5 individual samples.

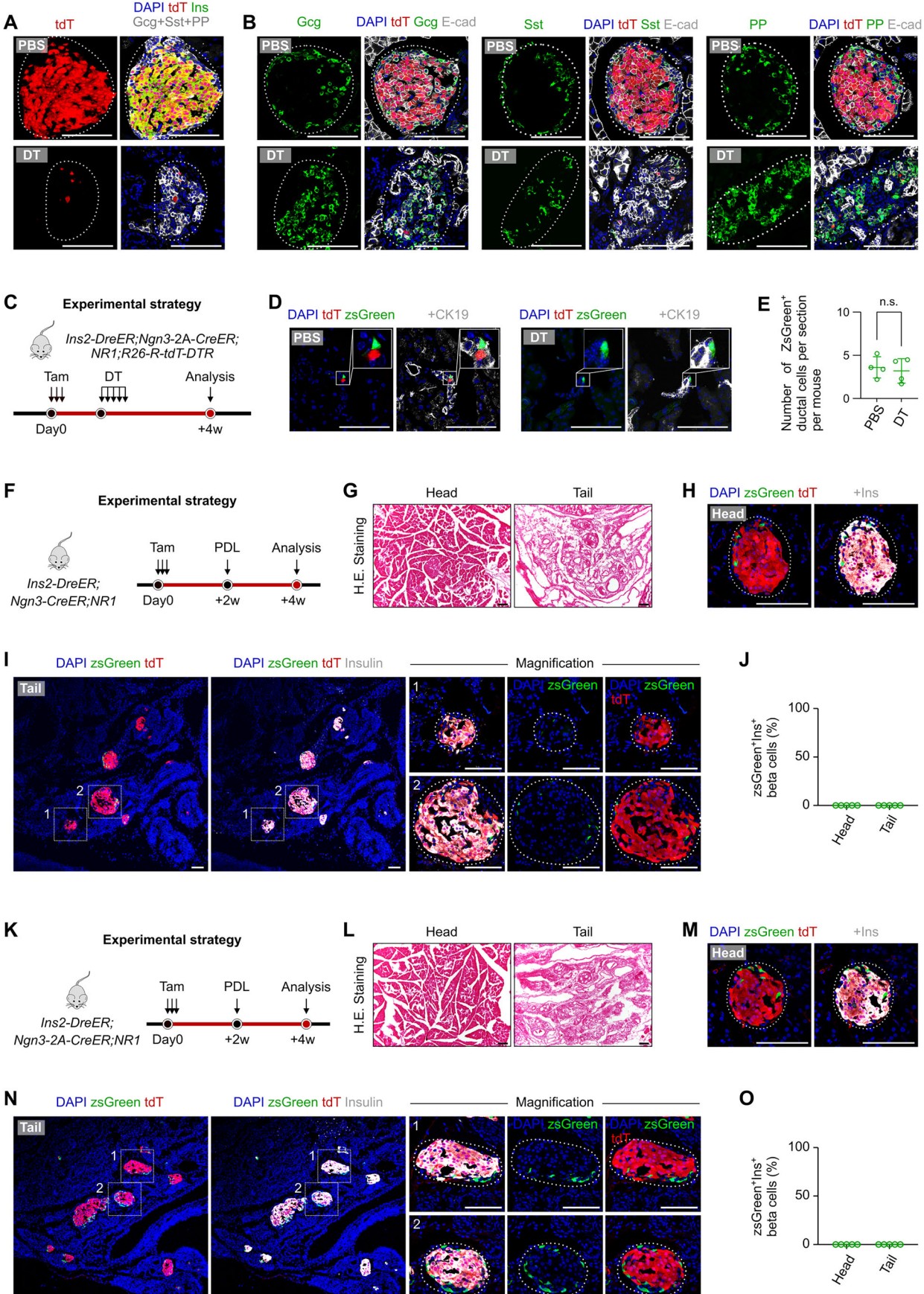

◀

**Figure EV4. Characterization of *Ins2-DreER;Ngn3-CreER;NR1* and *Ins2-DreER;Ngn3-2A-CreER;NR1* with PDL injury, related to Fig. 4.**

(A) Immunostaining for tdT, Ins, Gcg, Sst, and PP on pancreatic sections of *Ins2-DreER;R26-R-tdT*-DTR with PBS or DT treatment. (B) Immunostaining for tdT, Gcg, Sst, PP and E-cad on pancreatic sections of *Ins2-DreER;R26-R-tdT*-DTR with PBS or DT treatment. (C) Schematic showing the experimental strategy of *Ins2-DreER;Ngn3-2A-CreER;NR1;R26-R-tdT-DTR*. (D) Immunostaining for tdT, zsGreen and CK19 on pancreatic sections of *Ins2-DreER;Ngn3-2A-CreER;NR1;R26-R-tdT-DTR* mice with PBS or DT treatment. (E) Quantification of zsGreen$^+$ ductal cells per section of *Ins2-DreER;Ngn3-2A-CreER;NR1;R26-R-tdT-DTR* mice with PBS or DT treatment. Data are mean ± SD; PBS, $n = 4$ biological replicates; DT, $n = 4$ biological replicates; $P = 0.68$; n.s., non-significant. In each sample, islets from 10 pancreas sections were quantified. Two-tailed unpaired Student's $t$-tests were used for statistical comparisons and $p < 0.05$ was accepted as statistically significant. (F) Schematic showing the experimental strategy. (G) H.E. image of unligated head and ligated tail of the pancreas from *Ins2-DreER;Ngn3-CreER;NR1* mice 2 weeks after pancreatic ductal ligation (PDL) injury. (H) Immunostaining for tdT, zsGreen and Insulin on sections of unligated head of the pancreas from *Ins2-DreER;Ngn3-CreER;NR1* mice. (I) Immunostaining for tdT, zsGreen and Insulin on sections of ligated tail of the pancreas from *Ins2-DreER;Ngn3-CreER;NR1* mice. (J) Quantification of the percentages of ZsGreen$^+$ cells in Ins$^+$ pancreatic beta cells of unligated head and ligated tail of the pancreas from *Ins2-DreER;Ngn3-CreER;NR1* mice. Data are mean ± SD; $n = 5$ biological replicates. In each sample, islets from 10 pancreas sections were quantified. (K) Schematic showing the experimental strategy. (L) H.E. image of unligated head and ligated tail of the pancreas from *Ins2-DreER;Ngn3-2A-CreER;NR1* mice 2 weeks after PDL injury. (M) Immunostaining for tdT, zsGreen and Insulin on sections of unligated head of the pancreas from *Ins2-DreER;Ngn3-2A-CreER;NR1* mice. (N) Immunostaining for tdT, zsGreen and Insulin on sections of ligated tail of the pancreas from *Ins2-DreER;Ngn3-2A-CreER;NR1* mice. (O) Quantification of the percentages of ZsGreen$^+$ cells in Ins$^+$ pancreatic beta cells of unligated head and ligated tail of the pancreas from *Ins2-DreER;Ngn3-2A-CreER;NR1* mice. Data are mean ± SD; $n = 5$ biological replicates. In each sample, islets from 10 pancreas sections were quantified. Scale bars, 100 μm. Each image is representative of 4–5 individual mouse samples.

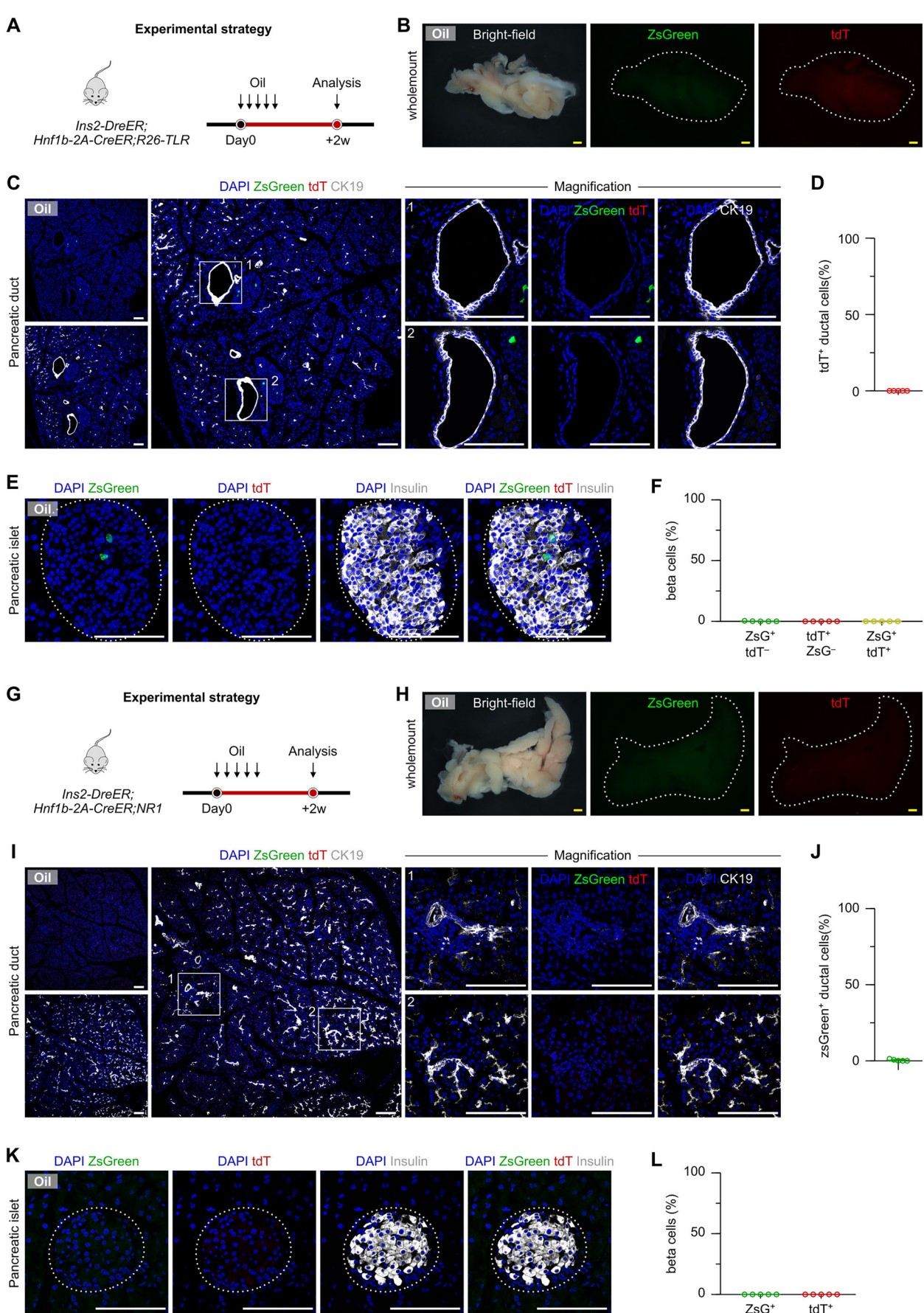

◄

**Figure EV5.  Characterization of *Ins2-DreER;Hnf1b-2A-CreER;R26-TLR* and *Ins2-DreER;Hnf1b-2A-CreER;NR1* without tamoxifen treatment, related to Figs. 5 and 6.**

(A) Schematic showing the experimental strategy of *Ins2-DreER;Hnf1b-2A-CreER;R26-TLR* with oil treatment. (B) Whole-mount fluorescent images of pancreas from *Ins2-DreER;Hnf1b-2A-CreER;R26-TLR* after oil treatment. (C) Immunostaining for ZsGreen, tdT and CK19 on pancreatic sections of *Ins2-DreER;Hnf1b-2A-CreER;R26-TLR* after oil treatment. (D) Quantification of the percentage of tdT$^+$ cells in CK19$^+$ pancreatic ductal cells of *Ins2-DreER;Hnf1b-2A-CreER;R26-TLR* after oil treatment. Data are mean ± SD; $n = 5$ biological replicates. In each sample, islets from 10 pancreas sections were quantified. (E) Immunostaining for ZsGreen, tdT and Ins on pancreatic sections of *Ins2-DreER;Hnf1b-2A-CreER;R26-TLR* after oil treatment. (F) Quantification of the percentage of ZsGreen$^+$tdT$^-$ or tdT$^+$ZsGreen$^-$ or ZsGreen$^+$tdT$^+$ cells in Ins$^+$ pancreatic beta cells of *Ins2-DreER;Hnf1b-2A-CreER;R26-TLR* after oil treatment. Data are mean ± SD; $n = 5$ biological replicates. In each sample, islets from 10 pancreas sections were quantified. (G) Schematic showing the experimental strategy of *Ins2-DreER;Hnf1b-2A-CreER;NR1* with oil treatment. (H) Whole-mount fluorescent images of pancreas from *Ins2-DreER;Hnf1b-2A-CreER;NR1* after oil treatment. (I) Immunostaining for ZsGreen, tdT and CK19 on pancreatic sections of *Ins2-DreER;Hnf1b-2A-CreER;R26-NR1* after oil treatment. (J) Quantification of the percentage of ZsGreen$^+$ cells in CK19$^+$ pancreatic ductal cells of *Ins2-DreER;Hnf1b-2A-CreER;R26-NR1* after oil treatment. Data are mean ± SD; $n = 5$ biological replicates. In each sample, islets from 10 pancreas sections were quantified. (K) Immunostaining for ZsGreen, tdT and Ins on pancreatic sections of *Ins2-DreER;Hnf1b-2A-CreER;R26-NR1* after oil treatment. (L) Quantification of the percentage of ZsGreen$^+$ or tdT$^+$ cells in Ins$^+$ pancreatic beta cells of *Ins2-DreER;Hnf1b-2A-CreER;R26-NR1* after oil treatment. Data are mean ± SD; $n = 5$ biological replicates. In each sample, islets from 10 pancreas sections were quantified. Scale bars, yellow, 1 mm; white, 100 µm. Each image is representative of 5 individual samples.

