## [Peer Review File · The EMBO Journal]

Ductal or Ngn3+ cells do not contribute to adult pancreatic islet beta-cell neogenesis in homeostasis

Bin Zhou, Xiuzhen Huang, Huan Zhao, Hui Chen, Zixin Liu, Kuo Liu, Zan Lv, Xiuxiu Liu, Ximeng Han, Maoying Han, Jie Lu, and Qiao Zhou

Corresponding author: Bin Zhou (zhoubin@sibs.ac.cn)

Review Timeline:

Submission Date:	17th Apr 24
Editorial Decision:	27th May 24
Revision Received:	17th Nov 24
Editorial Decision:	20th Dec 24
Revision Received:	10th Jan 25
Editorial Decision:	14th Mar 25
Revision Received:	15th Mar 25
Accepted:	21st Mar 25

Editor: Ieva Gailite

Transaction Report:

Dear Prof. Zhou,

Thank you for submitting your manuscript for consideration by the EMBO Journal. It has now been seen by three referees whose comments are shown below.

Given the referees' positive recommendations, I would like to invite you to submit a revised version of the manuscript, addressing the comments of all three reviewers. I should add that it is EMBO Journal policy to allow only a single round of revision, and acceptance of your manuscript will therefore depend on the completeness of your responses in this revised version.

Thank you for the opportunity to consider your work for publication. I look forward to your revision.

Yours sincerely,

Kelly M Anderson, PhD
Editor, The EMBO Journal
k.anderson@embojournal.org

We realize that it is difficult to revise to a specific deadline. In the interest of protecting the conceptual advance provided by the work, we recommend a revision within 3 months (25th Aug 2024). Please discuss the revision progress ahead of this time with the editor if you require more time to complete the revisions.

Referee #1:

The work by Zhao, Chen and colleagues "Intersectional lineage tracing reveals no contribution of ductal cells or Ngn3+ cells to islet beta cell neogenesis in adults" aims at solving the long-lasting controversy in the field of pancreas regeneration related to the existence and contribution of Ngn3+ ductal progenitors to islet beta cell reconstitution in homeostasis and after injury.

The Ngn3 expression in putative adult progenitors in mice has been a long-standing question and debate in the field that was re-activated by a recent Matters Arising (CellStemCell, 2023). Therefore, the acquisition of further evidence and clarification of the actual biological relevance of such adult ductal progenitor will be an important contribution allowing the field to move forward, especially if providing a definite answer of this debate.

I agree with the authors that many studies may have been misleading due to lack of careful controls related to possible off-target of lineage tracing systems used in the past as well as to low efficiency of Cre recombination with consequent suboptimal labelling. Therefore, the development of new strains and their careful evaluation may add additional evidence to support or confute the contribution of adult progenitors to beta cells in homeostasis.

The ms is clear and present complex transgenic mice exemplified with schematics of the experimental design which helps the reader through the several figure panels. I appreciate the use of different mouse models as well as the creation of two additional ones to cross-validate the results. The consistency of 2wks time point and 12wks chase period facilitate also the direct comparison, as well as the use of adult 8-10 wks old mice for all the experiments.

As mentioned above, a major question is the recombination efficiency and validation of the HNF1b-CreER that the submitted ms addresses: in the past the low recombination of HNF1b-CreER has been one of the major limitations to the acceptance of contribution of ductal cells to islets beta cells (Solar et al., DevCell 2009). The high efficiency in labelling the duct cells is an improvement over the previous papers.

Figure 4 presents a major problem: it is unclear what is the contribution of other islet-NON-beta-cells. What are the nuclei in figure 4D after DTR? They appear to be quite a lot in the islet regions that can still clearly identified. Could be that only 'degranulation' occurs? And therefore, restoration of INS+ cells may not be due to islet neo-genesis in this specific model? How is the potential contribution of duct cells excluded in this model?

Whole mount imaging is an improvement over section quantification in previous works. However, quantifications should be defined. How many islets/volumes/cells are counted per animal and per condition? Quantification details are missing. In general, the methods are described in a too basic way and, the quantifications must be reported in detail.

A few other issues remain unclear: what is the meaning of the 'very few' fluorescent cells without TAM administration? What is the nature of the Ngn3+ progenitors if not ductal cells? Islet progenitors? This point remains unclear. Would the conclusion on Ngn3-2A-CreER mice be that Ngn3 is expressed still by endocrine cells as well as ductal cells?

Finally, the Discussion should include a comprehensive interpretation of previous published data, including the (still) open question of the type of injury that may enhance and unveil the capacity of rare duct cells to contribute to beta cell regeneration. This is only partially evaluated in the current study. Furthermore, the statement 'we demonstrated that Dre/rox-mediated recombination effectively blocked the "unwanted" Cre reporter expression in beta cells, thereby facilitating the clear interpretation of lineage tracing data' must be supported by strong quantitative evidence.

In conclusion this is an important study that however requires significant revision.

It would be important to finally elucidate the biological significance of these rare putative ductal progenitors even assuming that they may exist. In fact, any transgenic model may have off target or other recombination problems and so far the debate is centred on the suitability of the models used.

Minor

-The terminology "super-efficient" could be changed to "highly efficient" and quantification clearly indicated.

-Figure legends should be expanded and described in more detail the results shown.

Referee #2:

This is an excellent paper that hopefully puts a nail in the coffin of the idea that duct or Ngn3+ cells in the adult contributes to beta cell formation under most circumstances in the adult animal. Numerous papers have been published in the past 20 years positing that Ngn3 or ductal cells can give rise to new beta cells during islet homeostasis or various forms of islet damage. Other papers have refuted the idea. As discussed in Ref. 20, anomalies with Cre drivers can confound lineage tracing. To the present authors' credit, they developed multiple new CreER drivers, in each case integrating CreER into the relevant loci (e.g. for Ngn3) instead of using a randomly inserted transgene. In addition, they performed double and in some cases triple lineage marking by combining different recombinases and reporter genes into a single strain. It is not easy to demonstrate a negative, but the authors did so as well as could be imagined. They also performed an extreme beta cell ablation model of Thorel et al (ref. 34) and showed that they could detect conversion from other endocrine cells in that instance, as a positive control. Finally they developed a new Hnf1b driven creER system which, combined with an Insulin2-DreER allele, demonstrated a lack of conversion of Hnf1b ductal cells to beta cells. Frankly I have no quibbles with the paper and commend the authors for laying out the diagrams and figures so nicely. I recommend the paper highly for EMBO J.

Referee #3:

This is a follow-up report of Magenheim et al. (Cell Stem Cell 2023) and Gribbn et al. (Cell Stem Cell 2021). The authors generated new knock-in mouse lines Ngn3-2A-CreER and Hnf1b-2A-CreER and used them as drivers to trace Ngn3+ cells or ductal cells. The authors successfully showed no de novo generation of insulin-producing beta cells from ductal cells or endogenous Ngn3-positive cells in the adult pancreas could be observed. The authors did an excellent job and showed solid evidence. This manuscript makes a valuable contribution to the field.

Response Letter to Reviewers

We thank the reviewers for all their valuable comments and constructive suggestions, which helped us significantly improve the quality of this work. The reviewers' constructive feedback was invaluable in refining the content of our manuscript. Below are the original reviewers' comments (in black) followed by our responses (in blue).

Referee #1:

The work by Zhao, Chen and colleagues "Intersectional lineage tracing reveals no contribution of ductal cells or Ngn3+ cells to islet beta cell neogenesis in adults" aims at solving the long-lasting controversy in the field of pancreas regeneration related to the existence and contribution of Ngn3+ ductal progenitors to islet beta cell reconstitution in homeostasis and after injury.

The Ngn3 expression in putative adult progenitors in mice has been a long-standing question and debate in the field that was re-activated by a recent Matters Arising (CellStemCell, 2023). Therefore, the acquisition of further evidence and clarification of the actual biological relevance of such adult ductal progenitor will be an important contribution allowing the field to move forward, especially if providing a definite answer of this debate.

I agree with the authors that many studies may have been misleading due to lack of careful controls related to possible off-target of lineage tracing systems used in the past as well as to low efficiency of Cre recombination with consequent suboptimal labelling. Therefore, the development of new strains and their careful evaluation may add additional evidence to support or confute the contribution of adult progenitors to beta cells in homeostasis.

The ms is clear and present complex transgenic mice exemplified with schematics of the experimental design which helps the reader through the several figure panels. I appreciate the use of different mouse models as well as the creation of two additional ones to cross-validate the results. The consistency of 2wks time point and 12wks chase period facilitate also the direct comparison, as well as the use of adult 8-10 wks old mice for all the experiments.

As mentioned above, a major question is the recombination efficiency and validation of the HNF1b-CreER that the submitted ms addresses: in the past the low recombination of HNF1b-CreER has been one of the major limitations to the acceptance of contribution of ductal cells to islets beta cells (Solar et al., DevCell 2009). The high efficiency in labelling the duct cells is an improvement over the previous papers.

We thank the reviewer for providing all valuable comments and constructive suggestions for improving the manuscript. The manuscript has been revised to incorporate new experiments and analyses, along with a comprehensive discussion, to strengthen our conclusions.

Figure 4 presents a major problem: it is unclear what is the contribution of other islet-NON-beta-cells. What are the nuclei in figure 4D after DTR? They appear to be quite a lot in the islet regions that can still clearly identified. Could be that only 'degranulation' occurs? And therefore, restoration of INS⁺ cells may not be due to islet neo-genesis in this specific model? How is the potential contribution of duct cells excluded in this model?

We thank the reviewer for raising this important question. To address this concern, we have conducted a more comprehensive analysis of the pancreas from *Ins2-DreER;R26-RSR-tdT-DTR* mice following either PBS or diphtheria toxin (DT) treatment (Figure EV4A). Specifically, we performed immunostaining for various endocrine cell markers within the islets, including glucagon (Gcg) for alpha cells, somatostatin (Sst) for delta cells, pancreatic polypeptide (PP) for PP cells, and insulin (Ins) for beta cells. As expected, in normal pancreatic tissues (PBS treatment), beta cells are the predominant endocrine cell type, clustering predominantly in the central region of the islets. In contrast, other endocrine cell types are localized primarily at the periphery of the islets. Notably, following DT treatment, there was a striking reduction in Ins⁺ beta cells. Concurrently, there was a notable increase in the population of non-beta endocrine cells, including alpha, delta, and PP cells. This result corroborates the successful ablation of beta cells by DT in our strategy. Furthermore, to corroborate the spatial distribution of these endocrine cell types within the islet regions, we performed immunostaining for E-cadherin (E-cad), a marker for epithelial cells, to indicate islet regions. By comparing the DT-treated group with the PBS-treated group, we confirmed that the number of alpha, delta, and PP cells in the islets increased, aligning with previous studies (Figure EV4B, Thorel et al, *Nature*, 2010, PMID: 20364121; Chera et al, *Nature*,

2014, PMID: 25141178; Zhao et al, *Nat Metab*, 2021, PMID: 33723463; Perez-Frances et al, *Nat Commun*, 2021, PMID: 34294685).

We contend that the cells that survived in the islet regions post-DT treatment are not primarily attributed to degranulation and transdifferentiation of preexisting beta cells into other endocrine cell types. The fundamental nature of the *Dre/rox*-mediated genetic recombination operates at the DNA level; thus, the genetic labeling with tdTomato of the vast majority of preexisting beta cells is permanent and irreversible. This marker persists regardless of whether these cells subsequently undergo proliferation or differentiation. If the surviving cells have originated from the degranulation of preexisting beta cells, even if they no longer express insulin, the genetic marker tdTomato would remain permanent and detectable. However, most cells that persist in the islet region are negative for tdT, indicating that degranulation is not the primary mechanism underlying their survival. Rather, alternative explanations, such as the differentiation of progenitor cells or the transdifferentiation of non-beta cells, may be more plausible explanations for this phenotype.

Previous studies have utilized lineage tracing techniques to illustrate the conversion of non-beta cells, such as alpha, delta, and PP cells, into insulin-producing cells following near-total beta-cell ablation. In 2010, Thorel et al. generated a *RIP-DTR* mouse model and achieved near-complete beta cell loss, resulting in the disappearance of 99.6% of beta cells. To ascertain the sources of regenerated beta cells, the researchers employed a mouse model *glucagon-rtTA;TetO-Cre;R26-LSL-YFP;RIP-DTR*. In this condition, doxycycline could induce alpha-cells labeled with YFP. Following DT-induced extreme beta cell ablation, they observed that the preexisting alpha-cells began expressing insulin (Thorel et al, *Nature*, 2010, PMID: 20364121). Similarly, in 2014 and 2021, they adopted analogous strategies and demonstrated the conversion of pancreatic delta-cells using *Sst-Cre;R26-LSL-YFP;RIP-DTR* (Chera et al, *Nature*, 2014, PMID: 25141178) and pancreatic polypeptide (PP)-expressing cells using *Ppy-rtTA;TetO-Cre;R26-LSL-YFP;RIP-DTR* (Perez-Frances et al, *Nat Commun*, 2021, PMID: 34294685) into insulin-producing cells after near-total beta-cell loss *in vivo*, respectively. These studies have elegantly demonstrated that the restoration of insulin-producing cells can occur via transdifferentiation of non-beta endocrine cell types in islets when beta cells are extremely deficient.

In our study, we employed a dual-recombinases-mediated intersectional genetic approach (*Ins2-DreER;Ngn3-2A-CreER;NR1;R26-R-tdT-DTR*) to investigate beta-cell neogenesis. This system permits the permanent and irreversible labeling of pancreatic pre-existing beta cells with

tdTomato and the diphtheria toxin receptor (DTR), whereas Ngn3⁺ non-beta cells are labeled with zsGreen upon tamoxifen (Tam) induction. Theoretically, pre-existing beta cells (tdTomato⁺) would be ablated following DT treatment, and the subsequent appearance of zsGreen⁺Ins⁺ cells indicates that they are derived from Ngn3⁺ non-beta cells. Indeed, we observed the reappearance of Ins⁺ cells at the chase time point, with some of these cells expressing zsGreen. This suggests that Ngn3⁺ non-beta cells can be converted to insulin-expressing cells following extreme beta-cell loss. This serves as a positive technical control for detecting beta cell generation from non-beta cells by using our system. However, a limitation of our approach is that it is unable to distinguish between Ngn3⁺ islet cells and Ngn3⁺ ductal cells as the exact source for newly generated insulin-expressing cells. This is because *Ngn3-2A-CreER* labels subpopulations of alpha-, delta-, and PP cells in the islets, as well as Ngn3-expressing cells located in the ducts, all of which are labeled by zsGreen. Based on previous studies as mentioned above, we postulated that these Ngn3⁺ non-beta cells responsible for the generation of Ins⁺ cells are likely the subpopulations of endocrine cells within the pancreatic islets. Further studies employing more precise lineage tracing strategies are necessary to elucidate the exact cellular origins of the newly generated Ins⁺ cells in this context.

We have included these new experimental data and interpretations, as well as the relevant references in the revised manuscript to provide a more comprehensive discussion of the findings. For detailed information, please refer to the revised Figure EV4 and Page 12, Line 20-28 of the manuscript.

Whole mount imaging is an improvement over section quantification in previous works. However, quantifications should be defined. How many islets/volumes/cells are counted per animal and per condition? Quantification details are missing. In general, the methods are described in a too basic way and, the quantifications must be reported in detail.

We thank the reviewer for the constructive suggestions, which were invaluable in improving the clarity of our work. We agree that detailed reporting of quantitative information is essential. To address the reviewer's concern, we have incorporated detailed quantitative information regarding the specific number of islet sections that were counted per mouse and per experimental condition, directly within the corresponding figure legend. We have also revised the quantification details in the Methods section of the manuscript for clarity. For details, please refer to the revised Figure legends and methods of the revised manuscript.

A few other issues remain unclear: what is the meaning of the 'very few' fluorescent cells without TAM administration? What is the nature of the Ngn3⁺ progenitors if not ductal cells? Islet progenitors? This point remains unclear. Would the conclusion on Ngn3-2A-CreER mice be that Ngn3 is expressed still by endocrine cells as well as ductal cells?

We thank the reviewer for the valuable comments. As a technical control, we have detected the fluorescent expression in islets without Tam administration. Regarding the ambiguity surrounding the term “very few” fluorescent cells without Tam administration, we have incorporated the specific quantification values in the revised manuscript, providing a clear understanding of the fluorescence expression in islets under non-induced conditions.

Our study showed that Ngn3-2A-CreER targeted endocrine cells as well as ductal cells (Figure 1M,N). Concerning the broad expression of Ngn3, we have adopted the dual-recombinase-mediated genetic approaches (for example, *Ins2-DreER;Ngn3-2A-CreER;R26-TLR* and *Ins2-DreER;Ngn3-2A-CreER;NRI*) to rigorously study the potential of Ngn3⁺ non-beta cells (including subpopulations of the alpha, delta, and PP cells within islets and Ngn3⁺ non-beta cells within ducts, as well as the putative Ngn3⁺ progenitors) to give rise to new beta cells under homeostatic conditions. Our system allowed for the permanent labeling of pre-existing beta cells and Ngn3⁺ non-beta cells upon Tam induction, ensuring that initial ectopic labeling of beta cells by *Ngn3-2A-CreER* was avoided or prevented. The lineage tracing results revealed that Ngn3⁺ non-beta cells do not make any detectable contribution to beta-cell neogenesis during homeostasis. These Ngn3⁺ non-beta cells, whether potential ductal or islet progenitors, do not contribute to beta cell neogenesis in the adult pancreas. This finding provides new insights to the ongoing discussion about the enigmatic nature of pancreatic progenitors and their role in beta-cell homeostasis and regeneration. We have included the above interpretations in the revised manuscript to provide a more comprehensive discussion. For details, please refer to Page 12, Line 4-16 of the revised manuscript.

Finally, the Discussion should include a comprehensive interpretation of previous published data, including the (still) open question of the type of injury that may enhance and unveil the capacity of rare duct cells to contribute to beta cell regeneration. This is only partially evaluated in the current study. Furthermore, the statement 'we demonstrated that Dre/rox-mediated recombination

effectively blocked the "unwanted" Cre reporter expression in beta cells, thereby facilitating the clear interpretation of lineage tracing data' must be supported by strong quantitative evidence.

We thank the reviewer for this important suggestion, which highlights the necessity for a more comprehensive interpretation of previous studies regarding the potential role of rare ductal cells, particularly Ngn3⁺ progenitors, in beta cell regeneration. To address this, we have expanded our Discussion section to provide a comprehensive review of the previous studies and the ongoing debate regarding the type of injury that may potentiate the regenerative capacity of these cells. Ngn3, a pivotal transcription factor in pancreatic development, is known for its central role in directing the endocrine fate of progenitor cells (PMID: 10677506). Genetic lineage tracing studies demonstrated the multipotent nature of Ngn3⁺ cells during embryogenesis, highlighting their remarkable plasticity (PMID: 11973276, 17609113). However, the precise identity and functional contribution of these cells in the adult pancreas remain unclear, with conflicting reports on their ability to contribute to beta-cell neogenesis. On the one hand, studies such as Xu et al.'s seminal work using a transgenic *Ngn3-nLacZ* model have suggested a significant contribution of adult pancreatic Ngn3⁺ progenitors to beta-cell regeneration following partial duct ligation (PDL) (PMID: 18243096). This finding generated considerable interest in the potential of adult Ngn3⁺ cells as therapeutic targets for diabetes. Nevertheless, subsequent investigations, notably Wang et al.'s utilizing Ngn3-knock-in mouse models, challenged this notion by demonstrating Ngn3 mRNA/protein expression within endocrine cells, including beta-cells, both in embryonic and adult stages (PMID: 19487660). Furthermore, studies from Kopp et al. and Xiao et al. have demonstrated PDL-induced Ngn3 expression in ductal cells but failed to conclusively establish beta-cell neogenesis from these progenitors (PMID: 21266405, 23867457). Recently, Gribben et al.'s identification of a subpopulation of Ngn3-expressing ductal progenitors capable of beta-cell generation under homeostatic conditions in the adult pancreas has reignited the debate on the existence and functionality of pancreatic stem cells (PMID: 34478642, 37028408, 37028401). Our study does not examine the potential of Ngn3⁺ ductal progenitor cells for beta-cell neogenesis under various injury models or some extreme conditions. However, our study highlights the urgent need for further research in this debated area. Our findings convincingly demonstrate that during homeostasis, Ngn3⁺ non-beta cells, which include the potential Ngn3⁺ ductal or islet progenitors, do not contribute to beta-cell neogenesis in the adult pancreas. Currently, it remains unclear which type of injury or extreme condition may unveil the regenerative capacity of rare Ngn3⁺ duct cells

to contribute to beta-cell regeneration. Accordingly, we have added the above interpretation in the Discussion section of the manuscript. For details, please refer to Page 12, Line 30, and Page 13, Line 1-7.

We agree with the reviewer's comment, emphasizing the need for robust quantitative evidence to support our claim that Dre/rox-mediated recombination effectively blocked the expression of the "unwanted" Cre reporter in beta cells, thereby enhancing the clarity and accuracy of lineage tracing data. To address this concern, we have included a thorough quantitative analysis, as follows. In light of the variability observed in previous studies, we hypothesize that discrepancies may arise from the expression of Ngn3 in a subpopulation of hormone-producing cells at adult stages, which could potentially confound the interpretation of lineage tracing results. Consequently, we recognized the necessity for a more precise genetic approach to minimize such confounding "unwanted" labeling issue. Initial experiments using the conventional lineage tracing system (*Ngn3-2A-CreER;R26-tdT*) demonstrated that about 16% of beta cells were labeled with tdT upon Tam treatment at the pulse stage (Fig.1K-M). However, the source of tdT⁺Ins⁺ cells at the chase stage remained unclear, as both Ngn3⁺ beta cells and Ngn3⁺ non-beta cells were pulse-labeled with tdT, demonstrating the limitations of the conventional Cre-loxP method. To address this issue, we used a dual recombinase-based strategy integrating Cre/loxP and Dre/rox recombination systems, thereby facilitating a more precise lineage tracing. We used the *R26-TLR* line, which features three distinct fluorescent reporters capable of simultaneously tracing Cre⁺Dre⁺, Cre⁻Dre⁺, and Cre⁺Dre⁻ cells within a single mouse. The use of *Ins2-DreER;Ngn3-CreER;R26-TLR* mice revealed that at the pulse stage, about 18% of beta cells were labeled (tdT⁺ZsGreen⁺), thereby distinctly segregating them from Ngn3⁺ non-beta cells (tdT⁺ZsGreen⁻). This permanent and irreversible labeling facilitated the unambiguous identification of the sources for newly generated beta cells. Furthermore, to independently examine whether Ngn3⁺ non-beta cells contribute to the beta cells in adults, we utilized a nested reporter system, *NRI*, where Cre-based lineage tracing reporter is conditionally controlled by Dre/rox recombination. In this design, *Ins2-DreER;Ngn3-CreER;NRI* mice were utilized to specifically label Ngn3⁺ non-beta cells with ZsGreen, while all beta cells, including those with Ngn3 expression, were marked with tdT. It is of particular importance to note that the Dre/rox-mediated recombination in beta cells ensured the excision of Cre-activated reporter genes, thus preventing the "unwanted" Cre reporter expression in beta cells. Quantitatively, about 0% of beta cells were labeled with ZsGreen (Cre/loxP reporter) at the pulse

stage, reinforcing the specificity of our approach. This design enabled accurate detection of even minimal contributions from Ngn3⁺ non-beta cells. The presented quantitative data support our statement that Dre/rox-mediated recombination effectively blocked the expression of the "unwanted" Cre reporter in beta cells. These data have been incorporated into the revised manuscript. For details, please refer to Page 4, Line 27-29, Page 6, Line 26-27, and Page 8, Line 1-2 of the revised manuscript.

In conclusion this is an important study that however requires significant revision.

It would be important to finally elucidate the biological significance of these rare putative ductal progenitors even assuming that they may exist. In fact, any transgenic model may have off target or other recombination problems and so far the debate is centred on the suitability of the models used.

We appreciated the reviewer for the insightful comments regarding the biological significance of the putative Ngn3⁺ ductal progenitors. Our current study, while unable to definitively confirm their biological significance, has made progress in elucidating their potential role through the development of new genetic tools and the dual-genetic tracing strategy during homeostasis. Specifically, two new mouse lines, *Ngn3-2A-CreER* and *Hnf1b-2A-CreER*, have been generated to minimize the potential effects of gene haploinsufficiency by preserving endogenous gene expression. Furthermore, using a dual-genetic tracing approach, we examined the contribution of Ngn3⁺ non-beta cells (including the potential Ngn3⁺ progenitors) and Hnf1b⁺ non-beta cells (including virtually all ductal cells) to beta-cell generation in adult homeostasis. Our findings convincingly demonstrated that neither population contributes to beta-cell renewal under homeostatic condition.

We acknowledge the inherent challenges in transgenic models, including potential off-target effects and ongoing debates regarding model suitability. In the case of the *Hnf1b-2A-CreER* line, although it labeled ductal cells with high efficiency, it also labeled a subset of islet endocrine cells. Therefore, using the dual-genetic tracing strategy based on the intersection of Ngn3 and Hnf1b (for example, *Ngn3-2A-CreER;Hnf1b-2A-DreER;R26-RSR-LSL-tdT*) cannot specifically target the potential Ngn3⁺ ductal progenitors. To overcome this limitation and to further explore the biology of these cells, we envision leveraging future advancements to identify more specific

markers for pancreatic ductal cells (e.g., X gene). With these new specific markers, we can generate more specific mouse models for targeting ductal cells (e.g., *X-2A-DreER*) and employ the dual-genetic tracing strategy (*Ngn3-2A-CreER;X-2A-DreER,R26-RSR-LSL-tdT*) to precisely trace the fate and functions of these rare Ngn3⁺ ductal progenitors. We have incorporated these considerations and future directions into our revised manuscript, please refer to the Page 13, Line 7-18 for detailed information.

Minor

-The terminology "super-efficient" could be changed to "highly efficient" and quantification clearly indicated.

We thank the reviewer for the valuable suggestion. We have replaced the term "super-efficient" with "highly efficient" throughout the manuscript. Furthermore, to ensure that our results are presented quantitatively, we have reviewed the relevant sections of the manuscript and added quantitative data.

-Figure legends should be expanded and described in more detail the results shown.

We thank the reviewer for raising this important point. Accordingly, we have revised the figure legends to provide a more detailed description of the results presented in each figure. For details, please refer to the revised figure legends in the manuscript.

Referee #2:

This is an excellent paper that hopefully puts a nail in the coffin of the idea that duct or Ngn3⁺ cells in the adult contributes to beta cell formation under most circumstances in the adult animal. Numerous papers have been published in the past 20 years positing that Ngn3 or ductal cells can give rise to new beta cells during islet homeostasis or various forms of islet damage. Other papers have refuted the idea. As discussed in Ref. 20, anomalies with Cre drivers can confound lineage tracing. To the present authors' credit, they developed multiple new CreER drivers, in each case integrating CreER into the relevant loci (e.g. for Ngn3) instead of using a randomly inserted transgene. In addition, they performed double and in some cases triple lineage marking by

combining different recombinases and reporter genes into a single strain. It is not easy to demonstrate a negative, but the authors did so as well as could be imagined. They also performed an extreme beta cell ablation model of Thorel et al (ref. 34) and showed that they could detect conversion from other endocrine cells in that instance, as a positive control. Finally they developed a new Hnf1b driven creER system which, combined with an Insulin2-DreER allele, demonstrated a lack of conversion of Hnf1b ductal cells to beta cells. Frankly I have no quibbles with the paper and commend the authors for laying out the diagrams and figures so nicely. I recommend the paper highly for EMBO J.

We thank the reviewer for the appreciation and constructive feedback on our work!

Referee #3:

This is a follow-up report of Magenheim et al. (Cell Stem Cell 2023) and Gribbn et al. (Cell Stem Cell 2021). The authors generated new knock-in mouse lines Ngn3-2A-CreER and Hnf1b-2A-CreER and used them as drivers to trace Ngn3+ cells or ductal cells. The authors successfully showed no de novo generation of insulin-producing beta cells from ductal cells or endogenous Ngn3-positive cells in the adult pancreas could be observed. The authors did an excellent job and showed solid evidence. This manuscript makes a valuable contribution to the field.

We thank the reviewer for appreciating our work!

Dear Bin,

Thank you for submitting your revised manuscript to The EMBO Journal. Your manuscript has now been seen by one of the original reviewers. As you will see, reviewer #1 has remaining concerns regarding data quantification. Furthermore, he/she is concerned with the shift in the main message of the manuscript and finds that a more definitive position would need to be taken by the study and recommends further analysis, which should provide stronger support for exclusion of ductal cell contribution to regenerating islet beta cells. Since the material required for this analysis would likely be already available, I would invite you to address these remaining points in a final revision round. If this analysis cannot be performed, please contact me to discuss the best way forward.

Furthermore, there are a few editorial and formatting aspects that would need to be addressed in the final version:

- 1) Please check that the funding information is correct and identical both in the manuscript and our online system. Currently, 2019YFA0802000 is not included in the Acknowledgments section.
- 2) CRediT has replaced the traditional author contributions section because it offers a systematic, machine-readable author contributions format that allows for more effective research assessment. Please remove the Authors Contributions from the manuscript and use the free text boxes beneath each contributing author's name in our online submission system to add specific details on the author's contribution. More information is available in our guide to authors.
- 3) Individual panels of Figures EV1 and EV4 are currently not mentioned in the manuscript text.
- 4) Our data editors have flagged the following issues in figure legends that need correcting:
 - Please indicate the statistical test used for data analysis in the legends of figures 1f, n; 2h, n; 3h, n; 5j; 6j.
 - Please define the white arrowheads in the legend of figure 1k, EV 2c-d; EV 3c-d.

We generally allow three months as standard revision time. Should you foresee a problem in meeting this deadline, please let us know in advance to discuss an extension. As a matter of policy, competing manuscripts published during this period will not negatively impact on our assessment of the conceptual advance presented by your study. However, please contact me as soon as possible upon publication of any related work to discuss the appropriate course of action.

When preparing your letter of response to the referees' comments, please bear in mind that this will form part of the Review Process File and will therefore be available online to the community. For more details on our Transparent Editorial Process, please visit our website: <https://www.embopress.org/page/journal/14602075/authorguide#transparentprocess>. Please also see the attached instructions for further guidelines on preparation of the revised manuscript.

Please feel free to contact me if have any further questions regarding this final revision. Thank you for the opportunity to consider your work for publication, and I look forward to discussing your revision with you.

With best wishes,

Ieva

We realize that it is difficult to revise to a specific deadline. In the interest of protecting the conceptual advance provided by the work, we recommend a revision within 3 months (20th Mar 2025). Please discuss the revision progress ahead of this time with the editor if you require more time to complete the revisions.

Referee #1:

The revised manuscript by Zhao, Chen and colleagues has addressed most of the experimental concerns raised in the initial review. Notably, the inclusion of the EV4 figure provides evidence confirming the nature of non-beta islet cells after DT treatment.

However, a significant issue remains regarding the detailed quantification of experimental data, both in the Methods section and figure legends. The total number of cells counted per section and per mouse replicate has not been specified, and the data are presented only as percentages. As requested previously, a table summarizing the raw numbers for each section used in the quantifications should be included or at least made available. Also whether a manual counting or any software has been used should be specified.

A more critical concern now refers to the revised key message of the paper. The original title, "Intersectional lineage tracing reveals no contribution of ductal cells or Ngn3⁺ cells to islet beta cell neogenesis in adults," has been changed to, "Minimal contribution of ductal cells or Ngn3⁺ cells to islet beta cell neogenesis in adults," determining a shift in the manuscript's conclusions and main message.

As stated in my initial review "The Ngn3 expression in putative adult progenitors in mice has been a long-standing question and debate in the field. Therefore, the acquisition of further evidence and clarification of the actual biological relevance of such adult ductal progenitors will be an important contribution allowing the field to move forward."

This manuscript could represent a significant advancement by providing a definitive answer to this debate. Unfortunately, the revised conclusions and added discussion appear to reignite uncertainty. Considering that a "gene X" enabling precise tracing of ductal cell progenitors has not been identified after nearly three decades of research and given the availability of extensive single-cell resolution data, it seems unlikely that this question will be resolved soon using genetic tools alone.

I am therefore unclear about how the authors have addressed the primary methodological criticisms, particularly the absence of necessary controls and the incomplete presentation of data.

I may propose a solution that, if supported by the data, could help clarify the study's conclusions and provide value to the field without bursting further controversy.

The current data seem to support the following conclusions:

1. There is no contribution of ductal progenitors to islet beta cell homeostasis in the adult mouse pancreas.
2. There is no available genetic tool that can definitively resolve whether rare Ngn3⁺ duct cells contribute to beta cell neogenesis after severe beta cell loss.

Nonetheless, the knock-in mouse models developed in this study present advantages over previous models. A potentially valuable experiment to assess whether Ngn3⁺ duct cells play a meaningful role following significant beta cell loss would involve:

- Quantifying the rare Ngn3⁺ duct cells before and after DT treatment in each section analyzed. Probably an extensive image analysis is required, which nowadays with automatation and clear staining could be performed more easily than in the past.
- Determining if there is a consistent absence of increased Ngn3⁺ cells in ducts while observing more Ngn3⁺ islet non-beta cells in each section after DT treatment.

If these analyses demonstrate no increase in Ngn3⁺ duct cells post-DT and show consistent localization of Ngn3⁺ islet non-beta cells, it will strongly suggest that rare duct cells are unlikely to migrate and proliferate significantly, even under severe beta cell loss.

This approach aligns with the need for rigorous and transparent quantification, which remains insufficiently addressed in the manuscript current form. By quantitatively supporting the improbability of rare duct cells having a meaningful biological role, this study could help resolve the long-standing debate and allow the field to move forward.

Response letter to Reviewer

We sincerely thank the reviewer for all valuable comments and constructive suggestions. These inputs have played an important role in improving the quality of our work. Below are the reviewer's original comments (shown in black) and our corresponding responses (shown in blue).

Referee #1:

The revised manuscript by Zhao, Chen and colleagues has addressed most of the experimental concerns raised in the initial review. Notably, the inclusion of the EV4 figure provides evidence confirming the nature of non-beta islet cells after DT treatment.

We thank the reviewer for the insightful review of our manuscript, which helped us to improve the quality of our work significantly.

However, a significant issue remains regarding the detailed quantification of experimental data, both in the Methods section and figure legends. The total number of cells counted per section and per mouse replicate has not been specified, and the data are presented only as percentages. As requested previously, a table summarizing the raw numbers for each section used in the quantifications should be included or at least made available. Also whether a manual counting or any software has been used should be specified.

We thank the reviewer for the important suggestion. Accordingly, we have included a detailed table (Appendix Table) in the revised manuscript, which summarizes the raw cell count numbers per section and per individual mouse used in the quantifications. We fully recognize the importance of providing comprehensive quantification details, including the specific quantification method. In the Methods section, we have stated that manual counting was used for cell quantification. For details, please refer to the Appendix Table, and Page 17, Line 5-7 of the revised manuscript.

A more critical concern now refers to the revised key message of the paper. The original title, "Intersectional lineage tracing reveals no contribution of ductal cells or Ngn3+ cells to islet beta cell neogenesis in adults," has been changed to, "Minimal contribution of ductal cells or Ngn3+ cells to islet beta cell neogenesis in adults," determining a shift in the manuscript's conclusions and main message.

We thank the reviewer for the insightful feedback and constructive suggestions, which have significantly contributed to enhancing our manuscript. We acknowledge the concern raised regarding the alteration of the paper's title from "Intersectional lineage tracing reveals no contribution of ductal cells or Ngn3⁺ cells to islet beta cell neogenesis in adults" to "Minimal contribution of ductal cells or Ngn3⁺ cells to islet beta cell neogenesis in adults". This modification was made with the intention of reflecting a more cautious interpretation of our experimental findings. There are long-standing uncertainties in the field regarding the role of ductal Ngn3⁺ progenitors in beta cell neogenesis. Using the intersectional lineage tracing genetic tools, our findings indicated that Ngn3⁺ non-beta cells do not contribute to the generation of pancreatic islet beta cells under homeostatic conditions. However, in the context of DT (diphtheria toxin)-induced severe beta cell loss, we observed the contribution from Ngn3⁺ non-beta cells to insulin-producing cells, which aligns with the previous observations reported by the Herrera group (PMID: 20364121, 25141178, 34294685). Given these observations, the title was initially revised, but we now realize that the revised title has reintroduced some ambiguity.

To address this issue accurately and unambiguously and in accordance with the journal's title length restrictions, the title was further revised to "Absence of ductal or Ngn3⁺ cell contribution to adult islet beta cell neogenesis in homeostasis". Concurrently, the abstract was revised, and the following statement was added: "Our data revealed no evidence of *de novo* generation of insulin-producing beta cells from ductal cells or endogenous Ngn3-positive non-beta cells in the adult pancreas during homeostasis. However, under extreme beta cell loss conditions in mice, Ngn3-positive non-beta cells could contribute to the generation of insulin-producing cells." We believe that this revision provides a clearer and more precise answer to the ongoing debate in the field and helps to resolve the long-standing uncertainties regarding the role of Ngn3⁺ progenitors in adult beta cell neogenesis. For details, please refer to the revised manuscript's Page 2, Line 12-14.

As stated in my initial review "The Ngn3 expression in putative adult progenitors in mice has been a long-standing question and debate in the field. Therefore, the acquisition of further evidence and clarification of the actual biological relevance of such adult ductal progenitors will be an important contribution allowing the field to move forward."

This manuscript could represent a significant advancement by providing a definitive answer to this debate. Unfortunately, the revised conclusions and added discussion appear to reignite uncertainty. Considering that a "gene X" enabling precise tracing of ductal cell progenitors has not been identified after nearly three decades of research and given the availability of extensive single-cell resolution data, it seems unlikely that this question will be resolved soon using genetic tools alone.

I am therefore unclear about how the authors have addressed the primary methodological criticisms, particularly the absence of necessary controls and the incomplete presentation of data.

I may propose a solution that, if supported by the data, could help clarify the study's conclusions and provide value to the field without bursting further controversy.

The current data seem to support the following conclusions:

1. There is no contribution of ductal progenitors to islet beta cell homeostasis in the adult mouse pancreas.
2. There is no available genetic tool that can definitively resolve whether rare Ngn3⁺ duct cells contribute to beta cell neogenesis after severe beta cell loss.

Nonetheless, the knock-in mouse models developed in this study present advantages over previous models. A potentially valuable experiment to assess whether Ngn3⁺ duct cells play a meaningful role following significant beta cell loss would involve:

- Quantifying the rare Ngn3⁺ duct cells before and after DT treatment in each section analyzed. Probably an extensive image analysis is required, which nowadays with automation and clear staining could be performed more easily than in the past.
- Determining if there is a consistent absence of increased Ngn3⁺ cells in ducts while observing more Ngn3⁺ islet non-beta cells in each section after DT treatment.

If these analyses demonstrate no increase in Ngn3⁺ duct cells post-DT and show consistent localization of Ngn3⁺ islet non-beta cells, it will strongly suggest that rare duct cells are unlikely to migrate and proliferate significantly, even under severe beta cell loss.

This approach aligns with the need for rigorous and transparent quantification, which remains insufficiently addressed in the manuscript current form. By quantitatively supporting the improbability of rare duct cells having a meaningful biological role, this study could help resolve the long-standing debate and allow the field to move forward.

We thank the reviewer for the valuable and constructive suggestions. We fully agree with the reviewer that the current data support the conclusion that “There is no contribution of ductal progenitors to islet beta cell homeostasis in the adult mouse pancreas.” Furthermore, we acknowledge the limitations imposed by the lack of a definitive genetic tool, such as "gene X", to precisely label and trace ductal Ngn3⁺ progenitors and agree that “There is no available genetic tool that can definitively resolve whether rare Ngn3⁺ duct cells contribute to beta cell neogenesis after severe beta cell loss.” We appreciate the reviewer’s proposed solution, which we believe has immense potential to further clarify the conclusions of our study.

Accordingly, we have quantified the number of the rare Ngn3⁺ duct cells in each analyzed section of *Ins2-DreER;Ngn3-2A-CreER;NRI;R26-R-tdT-DTR* mice, both with and without diphtheria toxin (DT) treatment. In this experimental design, Ins⁺ cells in both islets and ducts were labeled with tdT and expressed the diphtheria toxin receptor (DTR). Additionally, the Dre-rox recombination in pancreatic beta cells resulted in the excision of Cre-activated reporter genes and one loxP site, allowing only Ngn3⁺ non-beta cells in both islets and ducts to be labeled with zsGreen. Our results showed that with DT treatment, there were 3.20 ± 1.41 zsGreen⁺ ductal cells per section, compared to 3.60 ± 1.23 zsGreen⁺ ductal cells per section in the PBS-treated control group (Figure EV4C-E). Notably, we did not observe a significant increase in the number of Ngn3⁺ non-beta cells within the ducts following DT treatment (Figure EV4E). These results strongly suggest that these rare Ngn3⁺ non-beta cells within the ducts are highly unlikely to migrate and convert to islet beta cells, even under conditions of severe beta cell loss. We believe that this finding provides quantitative support that rare Ngn3⁺ duct cells are unlikely to play a meaningful biological role in islet beta cell neogenesis, thereby contributing to the resolution of a long-standing debate in the field.

We have included the new experimental data and interpretations in the revised manuscript to provide a more comprehensive discussion of the findings. For detailed information, please refer to the revised Figure EV4C-E and Page 9, Line 20-26 of the revised manuscript.

Once again, we thank the reviewer for the time and expertise in reviewing our manuscript!

Dear Bin,

Thank you for submitting a revised version of your manuscript to The EMBO Journal. Your manuscript has now been seen by one of the original reviewers. As you will see, reviewer #1 has remaining concerns regarding the role of duct-derived Ngn3+ cells upon severe injury.

After a discussion with the reviewer and yourself, as well as revisiting the comments of the other two reviewers during the first review round, I note that the other two reviewers do not find further extension of this aspect necessary for the publication of the study and its relevance for the research field. Therefore, I will accept your manuscript for publication after the final minor revision as outlined in our pre-decision discussion, in which you clarify the interpretation of the data and the rationale of the application of the injury conditions and include the negative data obtained on gentler injury conditions in the Appendix.

Please feel free to contact me if have any further questions regarding the revision. Thank you for the opportunity to consider your work for publication, and I look forward to receiving the final version of your manuscript.

With best wishes,

Ieva

We realize that it is difficult to revise to a specific deadline. In the interest of protecting the conceptual advance provided by the work, we recommend a revision within 3 months (12th Jun 2025). Please discuss the revision progress ahead of this time with the editor if you require more time to complete the revisions.

Referee #1:

The authors have partially addressed the central issue regarding the potential contribution of duct-derived Ngn3+ cells to beta cells following severe injury. Their new quantification indicates no increase in Ngn3+GFP non-beta cells within the ducts after DT treatment, suggesting they acknowledge this point. However, this important finding is not reflected in the title or abstract and is therefore not emphasised as a key conclusion of the manuscript.

Additionally, I remain concerned about the reliance on manual counting and the inconsistency in the conclusions. While the authors seem to agree with the previous comment and their data support the idea that duct-derived cells are unlikely to contribute significantly to beta cells after severe loss, their conclusions remain ambiguous.

Although the authors have added quantification of Ngn3+ non-beta cells in the ducts, the corresponding quantification of Ngn3+ non-beta cells in the islets after DT is still missing - although this was previously also requested. Figure EV4E and Appendix Table S10 should also include the number of Ngn3+ non-beta cells in the islets from the same sections and samples. Furthermore, any rare INS+GFP+ beta cells should be quantified within these same sections and samples.

Representative low-magnification images showing Ngn3+GFP+ non-beta cells in ducts along with islets should also be provided, as the current images primarily focus on islets. To improve clarity, the distribution of these ductal cells should be explicitly presented.

While Ngn3+ non-beta cells do contribute to beta cells following extensive beta-cell loss, it appears that a ductal origin is not the primary mechanism. This conclusion should be central to the manuscript and reflected in the main figures. If the authors are

unable to draw this conclusion, the impact of this work on resolving the ongoing debate in beta-cell biology remains limited. Given this, I question whether publishing these findings as they stand would contribute meaningfully to the field or merely prolong an already longstanding controversy.

The authors addressed the remaining editorial issues.

Dear Bin,

Thank you for implementing the final changes in the revised manuscript. I am now pleased to inform you that your manuscript has been accepted for publication in the EMBO Journal. Congratulations on a nice study!

I will now go through the synopsis text that you kindly provided and will let you know in the next couple of days whether any edits for style are needed.

Finally, we would like to promote your manuscript among the Chinese readership. Therefore, we would like to invite you to prepare a short summary of the manuscript in Chinese (1500-2000 Chinese characters), which we will promote on the WeChat platform "BioArt" more than 610,000 followers.

If you are interested in this opportunity, we recommend covering the article very close to its online publication date. Thus, ideally we will very much appreciate it if you can send us your summary within the next 7 working days. Please let us know whether or not you would be interested in contributing such a short summary in Chinese.

I have included below some general guidelines on how to prepare a summary and a link to a recent example for your reference. Please let me know if you have any questions about this.

If you have any questions, please do not hesitate to contact the Editorial Office. Thank you for your contribution to The EMBO Journal!

Best wishes,

Ieva

Ieva Gailite, PhD
Senior Scientific Editor
The EMBO Journal
Meyerohofstrasse 1
D-69117 Heidelberg
Tel: +4962218891309
i.gailite@embojournal.org

General Guidelines

1. These summary articles are meant to be targeting general audience so please limit the use of specialized technical terms, acronyms and jargon.
2. A summary usually starts with brief background information of the reported work, which is followed by explaining the findings in some detail, and ends with a short review of the conclusions as well as the implications of the work and future directions for the research.
3. The summary should contain a visual abstract, which can be the one provided in the paper.
4. Please provide ONE SINGLE document containing all text and graphical materials, ideally as a Word.docx or .doc file. Please DO NOT provide the document as a .pdf file.
5. Please DO NOT publicly release the document before the paper is officially published online.

Summary Example:

EMBO Journal | 灵珠与魔丸：昆虫miRNA调控病毒感染虫媒和植物的双重作用
